# Cohesin positions the epigenetic reader Phf2 within the genome

Wen Tang[1,2], Lorenzo Costantino[1,2], Roman Stocsits [1], Gordana Wutz[1], Rene Ladurner [1], Otto Hudecz [1], Karl Mechtler [1] & Jan-Michael Peters [1✉]

## Abstract

**Genomic DNA is assembled into chromatin by histones, and extruded into loops by cohesin. These mechanisms control important genomic functions, but whether histones and cohesin cooperate in genome regulation is poorly understood. Here we identify Phf2, a member of the Jumonji-C family of histone demethylases, as a cohesin-interacting protein. Phf2 binds to H3K4me3 nucleosomes at active transcription start sites (TSSs), but also co-localizes with cohesin. Cohesin depletion reduces Phf2 binding at sites lacking H3K4me3, and depletion of Wapl and CTCF repositions Phf2 together with cohesin in the genome, resulting in the accumulation of both proteins in chromosomal regions called vermicelli and cohesin islands. Conversely, Phf2 depletion reduces cohesin binding at TSSs lacking CTCF and decreases the number of short cohesin loops, while increasing the length of heterochromatic B compartments. These results suggest that Phf2 is an 'epigenetic reader', which is translocated through the genome by cohesin-mediated DNA loop extrusion, and which recruits cohesin to active TSSs and limits the size of B compartments. These findings reveal an unexpected degree of cooperativity between epigenetic and architectural mechanisms of eukaryotic genome regulation.**

**Keywords** Chromatin; Compartments; DNA Loop Extrusion; Epigenetics; SMC Complexes
**Subject Category** Chromatin, Transcription & Genomics

## Introduction

In eukaryotes, genomic DNA is wrapped around histone octamers to form nucleosomes, which assemble into chromatin fibers (Kornberg, 1974; Olins and Olins, 1974). The properties of chromatin are epigenetically controlled by histone modifying enzymes, which generate posttranslational modifications that correlate with gene expression states (Allis and Jenuwein, 2016; Zhao and Garcia, 2015). Histone modifications are recognized by epigenetic reader proteins that can further modify the properties of

chromatin. For example, in euchromatic regions, the TSSs of active genes are flanked by nucleosomes that are tri-methylated on lysine 4 of histone H3 (H3K4me3) (Bernstein et al, 2005; Santos-Rosa et al, 2002; Wang and Helin, 2024). These H3K4me3 marks are recognized by 'plant homeodomain (PHD) finger' domains in nucleosome remodelers and transcriptional regulators (Becker, 2006; Li et al, 2006; Pena et al, 2006; Shi et al, 2006; Wysocka et al, 2006). In contrast, nucleosomes in heterochromatic regions are modified by mono-, di-, and tri-methylation of lysine 9 of histone H3 (H3K9me1/2/3). These marks recruit heterochromatin protein 1 (HP1) (Bannister et al, 2001; Lachner et al, 2001), and thereby repress gene expression (Montavon et al, 2021; Padeken et al, 2022). Histone demethylases of the Jumonji C (JmjC) family can remove both euchromatic and heterochromatic methyl-marks, thus preventing their interactions with reader proteins and changing gene expression and cell fate during differentiation and diseases (Kooistra et al, 2012; Shi and Whetstine, 2007).

In addition to being epigenetically controlled, chromatin fibers are spatially regulated through the formation of loops, which can span hundreds of kilobases in length (Rao et al, 2014). Many chromatin loops are generated by cohesin (Gassler et al, 2017; Rao et al, 2017; Schwarzer et al, 2017; Wutz et al, 2017), a 'structural maintenance of chromosomes' (SMC) complex, which extrudes DNA into loops in vitro (Davidson et al, 2019; Kim et al, 2019). In cells, cohesin loops are typically formed within the boundaries of topologically associating domains (TADs) (Dixon et al, 2012; Nora et al, 2012). Most TAD boundaries depend on the DNA binding protein CTCF (Nora et al, 2017; Wutz et al, 2017), which stops cohesin-mediated loop extrusion (Davidson et al, 2023; Zhang et al, 2023), but the replicative MCM helicase (Dequeker et al, 2022) and active genes (Banigan et al, 2023) are also cohesin boundaries. Cohesin loops can mediate long-range enhancer-promoter interactions (Cheng et al, 2023; El Khattabi et al, 2019; Kiefer et al, 2023; Kiefer et al, 2024; Thiecke et al, 2020) and gene recombination (Hill et al, 2020; Zhang et al, 2019).

The length and lifetime of cohesin loops are limited by Wapl, which releases cohesin from chromatin (Kueng et al, 2006; Tedeschi et al, 2013) and thus dissolves cohesin loops. In Wapl-depleted cells, the prolonged residence time of cohesin on chromatin results in the formation of longer cohesin loops (Haarhuis et al, 2017; Wutz et al, 2017) and the accumulation of cohesin in axial chromosomal regions called 'vermicelli' (Tedeschi

[1]Research Institute of Molecular Pathology (IMP), Vienna Biocenter (VBC), Campus-Vienna-Biocenter 1, 1030 Vienna, Austria. [2]These authors contributed equally: Wen Tang, Lorenzo Costantino. ✉E-mail: peters@imp.ac.at

et al, 2013). If, in addition to Wapl, CTCF is also depleted, loop extruding cohesin complexes accumulate in 'cohesin islands', which span several kilobases in length and are often located at sites of convergent transcription (Busslinger et al, 2017). Downregulation of Wapl and CTCF is also used under physiological conditions to regulate cohesin-mediated loop extrusion. At the *Igf2-H19* locus, CTCF binding is regulated by differential DNA methylation to allow the formation of allele-specific cohesin loops that control enhancer-promoter interactions (Kurukuti et al, 2006; Nativio et al, 2009; Splinter et al, 2006), and in pro-B cells and developing olfactory neurons Wapl transcription is downregulated to enable the formation of long cohesin loops for V(D)J recombination (Hill et al, 2020; Zhang et al, 2019) and stochastic protocadherin promoter choice, respectively (Kiefer et al, 2023; Kiefer et al, 2024).

Except for the spreading of DNA damage induced histone phosphorylation (Arnould et al, 2021), cohesin and histone modifying enzymes are thought to exert their regulatory functions independently of each other. However, here we show that cohesin directly interacts with Phf2, an epigenetic reader protein and a member of the JmjC family of histone demethylases, suggesting direct interactions between architectural and epigenetic mechanisms of genome regulation. During the preparation of this manuscript, cohesin was conversely also identified as an interactor of Phf2 (Feng et al, 2024).

Phf2 mRNA is widely expressed in mammalian tissues (Hasenpusch-Theil et al, 1999). Phf2 has a N-terminal PHD domain that binds with nanomolar affinity to H3K4me3 and H3K4me2 peptides in vitro (Horton et al, 2023; Wen et al, 2010). Phf2 has also been identified as a H3K4me3 interacting protein by mass spectrometry (Bluhm et al, 2016; Eberl et al, 2013) and co-localizes with the H3K4me3 mark in the vicinity of active TSSs in chromatin immunoprecipitation-sequencing (ChIP-seq) experiments (Bricambert et al, 2018; Pappa et al, 2019). Phf2's JmjC domain is unusual because a conserved histidine residue, which helps to coordinate a catalytically important Fe(II) ligand in other JmjC domain proteins, is replaced by a tyrosine residue (Horton et al, 2011). Phf2 shares this property with *S. pombe* Epe1, which regulates the genomic distribution of H3K9me3 in vivo but has no detectable histone demethylase activity in vitro (Ayoub et al, 2003; Braun et al, 2011; Ragunathan et al, 2015; Trewick et al, 2007). In contrast, Phf2 has been reported to demethylate H3K9me2 histones and nucleosomes in vitro (Baba et al, 2011; Bricambert et al, 2018; Horton et al, 2023).

Phf2 knockout mice are viable until birth (Okuno et al, 2013), perhaps because Phf2 functions redundantly with the related enzymes Phf8 and Kdm7a (Wen et al, 2010). However, Phf2-null mice suffer from growth retardation, reduced adipose tissue, and often die postnatally (Okuno et al, 2013). Phf2 has also been implicated in controlling transcription of ribosomal genes (Shi et al, 2014) and in gene regulation, differentiation and proliferation of chondrocytes (Hata et al, 2013), osteoblasts (Kim et al, 2014), hepatocytes (Bricambert et al, 2018; Lane et al, 2019), neural progenitors (Aguirre et al, 2024; Pappa et al, 2019) and macrophages (Stender et al, 2012).

Here we show that interactions between cohesin and Phf2 contribute to the genomic distribution of Phf2, possibly because Phf2 'travels' with loop extruding cohesin, that Phf2 limits the size of heterochromatic B compartments, and that Phf2 helps to anchor loops formed by cohesin at active TSSs lacking CTCF.

# Results

## Identification of Phf2 as a cohesin interacting protein

CTCF stops loop-extruding cohesin in part by binding via a YxF motif to a 'conserved essential surface' (CES) on cohesin (Li et al, 2020). However, this micromolar interaction is difficult to maintain during protein purification (Stedman et al, 2008; Wendt et al, 2008). CTCF's role as a cohesin boundary was therefore discovered by the co-localization of cohesin and CTCF (Parelho et al, 2008; Stedman et al, 2008; Wendt et al, 2008) at TAD boundaries (Dixon et al, 2012; Fudenberg et al, 2016; Nora et al, 2012; Sanborn et al, 2015) and not by their physical interaction. We therefore reasoned that other cohesin regulators could also have been missed by purification approaches. Consistent with this possibility, several proteins other than cohesin and CTCF are enriched at TAD boundaries (Dataset EV1).

We therefore established formaldehyde crosslinking conditions, which stabilize cohesin-CTCF interactions during cohesin affinity purification (Appendix Fig. S1A,B). To identify specific cohesin interactors in such samples, we searched for proteins that co-purify with chromatin-bound cohesin more abundantly following Wapl depletion, which increases cohesin levels on chromatin and leads to the accumulation of cohesin at CTCF sites (Haarhuis et al, 2017; Kueng et al, 2006; Tedeschi et al, 2013; Wutz et al, 2017). Quantitative mass spectrometry revealed that the topoisomerase Top2b, the bromodomain protein Brwd1 and the PHD finger protein Phf2 were highly enriched in cohesin isolates from Wapl depleted mouse embryonic fibroblasts (MEFs), generated by Cre recombinase-mediated deletion of floxed *Wapl* alleles, hereafter *Wapl* KO (Tedeschi et al, 2013), compared to cohesin isolates from MEFs expressing Wapl, hereafter *Wapl* WT (Fig. 1A; Appendix Fig. S1C and Dataset EV2). As expected, Wapl was depleted from cohesin samples isolated from *Wapl* KO MEFs. MCM helicase subunits were also reduced, possibly because MCM levels were lower in these cells (Appendix Fig. S1D) but also consistent with the possibility that Wapl depletion relocates cohesin from MCM to CTCF boundaries (Dequeker et al, 2022).

To further investigate whether cohesin interacts with Top2b, Brwd1, and Phf2, we examined their localization in Wapl-depleted MEFs. In these cells, cohesin accumulates in axial chromosomal vermicelli (Tedeschi et al, 2013), which represent the bases of long cohesin loops (Fudenberg et al, 2016; Haarhuis et al, 2017; Wutz et al, 2017). Fluorescence microscopy revealed that Phf2 co-localized with cohesin in vermicelli in fixed and live *Wapl* KO MEFs (Fig. 1B,C; Appendix Fig. S1E), but this was not the case for Top2b, Brwd1 (Appendix Fig. S1F) and other epigenetic reader proteins, which were not enriched in cohesin samples (Baz1b, Chd4, the Phf2 paralog Phf8; Appendix Fig. S1G,H). However, during this study Top2b was reported to co-localize with cohesin at TAD boundaries (Uuskula-Reimand et al, 2016) and Brwd1 to regulate cohesin dynamics on chromatin (Mandal et al, 2024). Top2b and Brwd1 may therefore be cohesin interactors that are not translocating with cohesin into vermicelli or might be too abundant in other nuclear locations to become detectably enriched in vermicelli. We therefore focused on characterizing interactions of cohesin with Phf2.

## Phf2 binds to the kleisin-STAG module of cohesin

Wapl depletion did not only enrich Phf2 in cohesin samples, but also cohesin in Phf2-GFP immunoprecipitates (Fig. 1D; Appendix

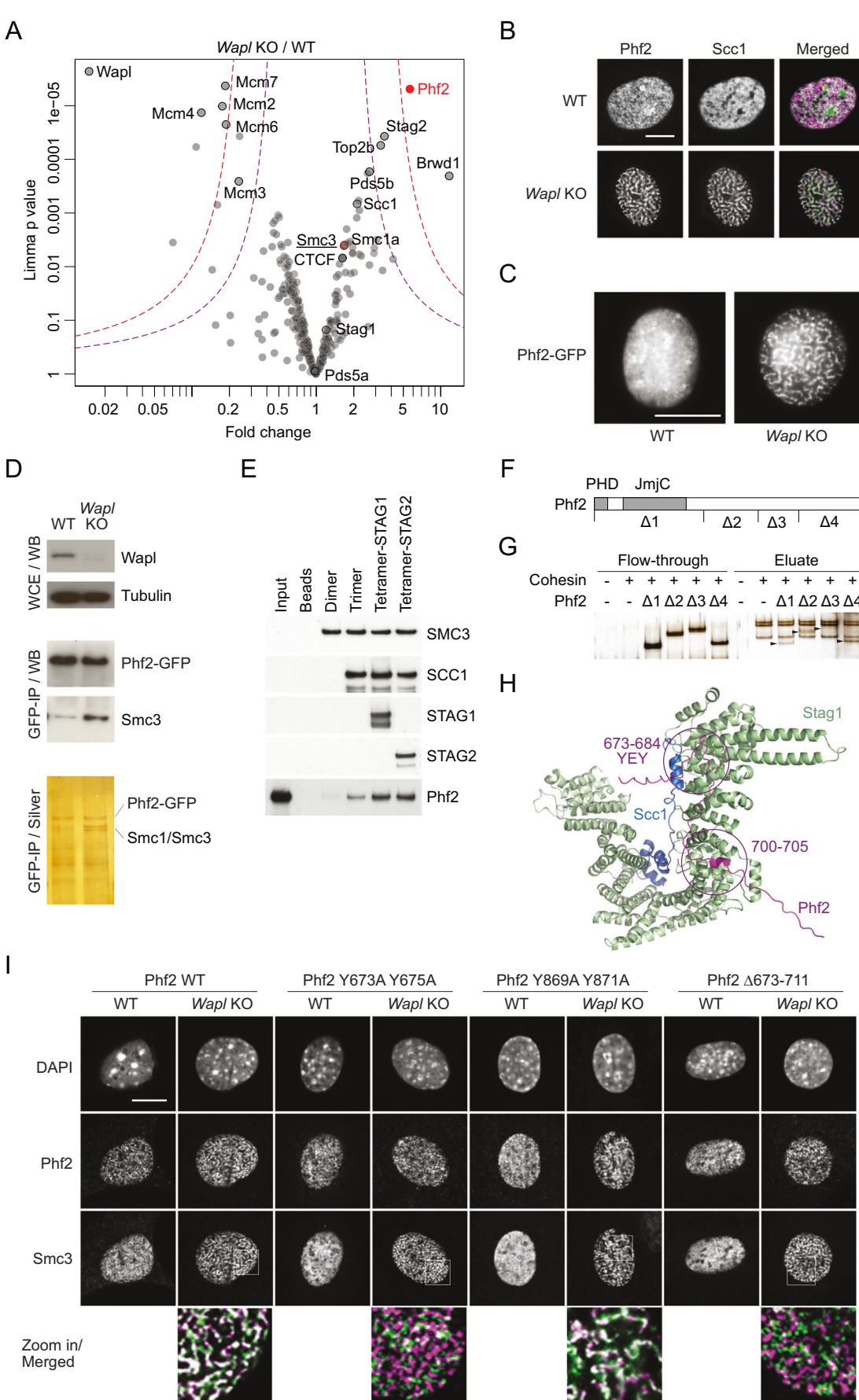

◄ **Figure 1.  Identification of Phf2 as a cohesin interacting protein.**

(A) Volcano plot of proteins identified by Smc3 ChIP-qMS, normalized to total protein in cohesin immunoprecipitates from WT versus *Wapl* KO MEFs. The plot shows enriched proteins in the *Wapl* KO sample on the right and depleted proteins on the left. Biological replicates $n = 3$. Statistical significance of differentially expressed proteins was determined using limma. Proteins above the purple and red dotted lines represent significance thresholds of $p < 0.05$ and $p < 0.01$, respectively. The top enriched protein, Phf2, is highlighted in red. (B) Fluorescence microscopy with Phf2 and Scc1 antibodies in WT and *Wapl* KO MEFs. In the merged panel, Phf2 is shown in green and Scc1 in magenta. Scale bar, 10 μm. (C) Live cell microscopy of Phf2-GFP in WT and *Wapl* KO MEFs. Scale bar, 10 μm. (D) Top panel: Immunoblot analysis of whole-cell extracts (WCE) from WT and *Wapl* KO MEFs. Middle panel: Immunoblot analysis of samples immunoprecipitated (IP) using GFP antibody from WT and *Wapl* KO MEFs expressing Phf2-GFP. Bottom panel: Silver staining of P samples as described in the middle panel. (E) Immunoblot analysis of in vitro binding assay of Phf2 with the cohesin complex. Human recombinant cohesin complexes (Dimer, Trimer, or Tetramer) bound to antibody beads were mixed with purified Phf2. Bound proteins were analyzed using the indicated antibodies. (F) Schematic representations of full-length Phf2 and its deletion mutants. PHD, plant homeodomain; JmjC, Jumonji C domain. Δ1:1–450aa; Δ2:451–659aa; Δ3:660–819aa; Δ4:820–1096aa. (G) Silver staining of in vitro binding assay of Phf2 deletion mutants with human cohesin. Cohesin tetramer-STAG1 was mixed with purified Phf2 mutants (Δ1, Δ2, Δ3, or Δ4). Flow-through and eluate were analyzed, and the arrowheads marked the positions of the corresponding Phf2 mutants. (H) AlphaFold2 model for the interaction between Scc1/Stag1 and Phf2. (I) Fluorescence microscopy with Phf2 and Smc3 antibodies in WT and *Wapl* KO MEF using Phf2 WT, Phf2 Y673A Y675A, Phf2 Y869A Y871A, and Phf2 Δ673–711. In the merged magnified panels, Phf2 is shown in green and Smc3 in magenta. Scale bar, 10 μm. Source data are available online for this figure.

Fig. S2A,B), further indicating that cohesin-Phf2 interactions are specific. In vitro, purified Phf2 (Appendix Fig. S2C) bound to human cohesin trimers composed of SMC1, SMC3, and the kleisin SCC1 (also known as RAD21 and Mcd1), and more abundantly to tetramers that also included STAG1 or STAG2, but not to SMC1-SMC3 dimers (Fig. 1E; Appendix Fig. S2D). These results indicate that Phf2 interacts directly with cohesin by binding to its kleisin-STAG module, as does CTCF (Li et al, 2020).

Deletion analysis showed that Phf2's PHD finger and JmjC domains are not required for cohesin binding in vitro, whereas amino acid residues 660–819 of Phf2 are (Fig. 1F,G; Appendix Fig. S2E,F; Phf2-Δ3). Expression of these mutants in immortalized MEFs (iMEFs) revealed that residues 660–819 were also required for enrichment of Phf2 in vermicelli, whereas other Phf2 domains were not (Appendix Fig. S2G,H), indicating that this region of Phf2 is also required for cohesin binding in cells. In this part of the Phf2 sequence, αFold2 predicted residues 673–684 and 700–705 to interact with Stag1-Scc1 (Fig. 1H). Interestingly, residues 673–684 contain an evolutionarily conserved YEY motif (residues 673–675) that is predicted to bind to the same CES on Stag1-Scc1 as CTCF's YxF motif, although in a different orientation (Appendix Fig. S2I,J).

To test whether Phf2's YEY motif is required for cohesin binding we mutated the two tyrosine residues to alanine, expressed the resulting Phf2 Y673A Y675A mutant in MEFs, depleted Wapl and analyzed whether Phf2 still co-localizes with the cohesin subunit Smc3 in vermicelli. We also generated Phf2 mutants in which we either deleted 39 residues including the YEY motif (Phf2 Δ673–711) or mutated the tyrosine residues in a similar YVY motif (Phf2 Y869A Y871A), which is located in an unstructured region that does not bind cohesin (Δ4 in Fig. 1F,G). The Phf2 YEY motif and Phf2 Δ673–711 mutants were greatly reduced in their co-localization with cohesin in vermicelli, whereas mutation of the YVY motif did not have such an effect (Fig. 1I). These results indicate that Phf2's YEY motif is required for cohesin binding, possibly because it binds the same CES as CTCF.

## Phf2 co-localizes with H3K4me3 nucleosomes and cohesin in the mouse genome

In previous ChIP-seq experiments, Phf2 was found to co-localize with H3K4me3-nucleosomes (Bricambert et al, 2018; Pappa et al, 2019), consistent with its ability to bind H3K4me3 peptides in vitro (Bricambert et al, 2018; Horton et al, 2011; Horton et al, 2023; Wen

et al, 2010) and to co-purify with H3K4me3 nucleosomes from cells (Bluhm et al, 2016; Eberl et al, 2013). However, our findings that Phf2 accumulates with cohesin in vermicelli and binds cohesin in vitro predicted that the distribution of Phf2 in the genome should also overlap with that of cohesin. We, therefore, re-assessed the genomic distribution of Phf2 by ChIP-seq in MEFs.

These experiments identified 21,326 Phf2 peaks in the mouse genome, most of which overlapped with H3K4me3-nucleosomes (17,064, 80%; Fig. 2A–C). As expected, since H3K4me3-nucleosomes are located at active TSSs, these Phf2 peaks also overlapped with TSSs (Appendix Fig. S3A,B) and RNA polymerase II phosphorylated on serine residue 5 (PolII Ser5-P; Fig. 2C). Pile-up analyses showed that Phf2 and H3K4me3 signals peaked directly in front and behind TSSs, presumably reflecting nucleosomes at −1 and +1 positions (Appendix Fig. S3C). The Phf2 peaks that did not overlap with H3K4me3 sites (4262, 20%) were smaller than those at H3K4me3 sites (Fig. 2A,C). These small peaks were nevertheless also specific for Phf2, as almost all Phf2 peaks (99.3%) were absent in *Phf2*^t/f^ MEFs following tamoxifen-induced expression of ER-Cre and subsequent Phf2 depletion (hereafter referred to as *Phf2* KO; Figs. 2A,D,E and EV1A). These results confirm that Phf2 is an epigenetic reader protein that co-localizes in many cases with H3K4me3-nucleosomes in the vicinity of active TSSs.

However, 12,198 Phf2 peaks (57%) also overlapped with peaks of the cohesin subunit Smc3 (Fig. 2A,F,G) and 6955 with CTCF (33%; Figs. 2A and EV1B,C). Pile-up analyses revealed that some Smc3 signal enrichment could be detected at most Phf2 peaks and conversely some Phf2 at most Smc3 peaks, independent of whether these co-localized with H3K4me3 or not (Fig. 2G). The overlap between Phf2 and cohesin peaks therefore underestimates the extent to which these proteins co-localize in the genome, due to the thresholding used during ChIP-seq peak calling. In contrast, CTCF was only detected at a subset of Phf2 peaks (Figs. 2A and EV1B,C). Pile-up analyses revealed that the peak of Phf2 ChIP-seq signals was closer to the peak of cohesin signals than to the peak of CTCF, suggesting that Phf2 accumulates at some CTCF sites because Phf2 is bound to cohesin (Fig. EV1D).

To test whether the enrichment of Phf2 at Smc3 sites depends on cohesin, we analyzed the distribution of Phf2 in *Smc3* KO MEFs (Busslinger et al, 2017). Western blot analyses indicated that the levels of Phf2 were reduced by ~30% in *Smc3* KO MEFs but only by ~13% in chromatin fractions obtained from these cells (Figs. 3A and EV2A,B). ChIP-seq experiments indicated that Smc3 depletion also reduced the

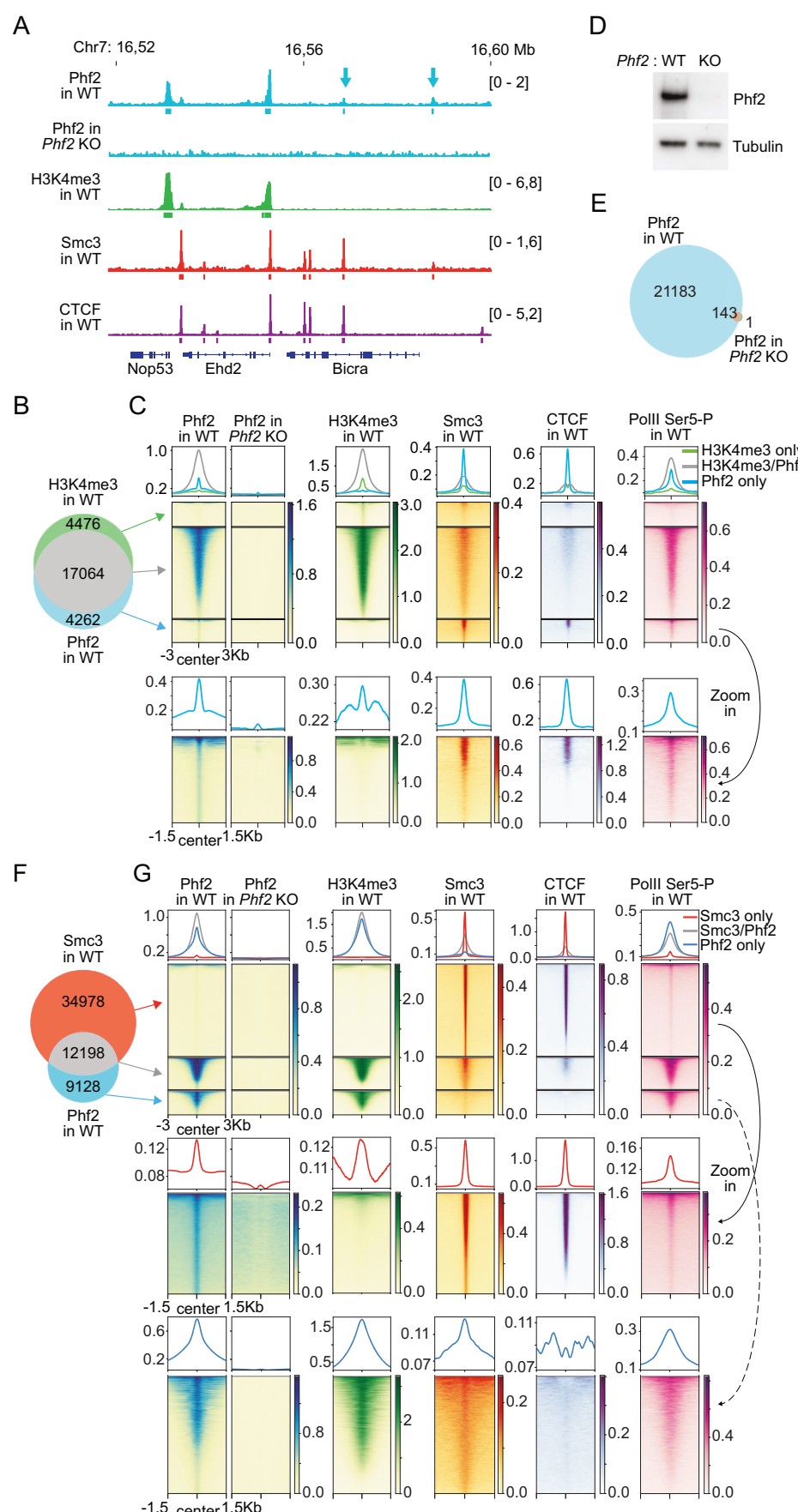

**Figure 2. Phf2 colocalizes with H3K4me3 and cohesin genome-wide.**

large Phf2 peaks that overlapped with H3K4me3 by ~13%. However, small Phf2 peaks that overlapped with cohesin but not with H3K4me3 were reduced by 64% in *Smc3* KO MEFs, and accordingly 5843 peaks could no longer be detected in these cells (27% of all Phf2 peaks in WT MEFs; Figs. 3B–D and EV2C,D). The finding that small Phf2 peaks are relatively more affected than large peaks by Smc3 depletion suggests that small Phf2 peaks become undetectable in *Smc3* KO MEFs not only because total Phf2 levels are reduced in these cells but also because cohesin is required to recruit Phf2 to these sites. These results suggest that cohesin is required for the positioning of Phf2 at 'cohesin-only' sites but that cohesin is dispensable for enrichment of Phf2 at H3K4me3-nucleosomes.

## Cohesin influences the chromatin association of Phf2

Because Phf2 interacts with cohesin and both proteins accumulate in vermicelli following Wapl depletion, we performed fluorescence recovery after photobleaching (FRAP) experiments to assess how the depletion of cohesin and Wapl affects the interaction of Phf2 with chromatin. These experiments revealed that in MEFs, 35% of ectopically expressed Phf2-GFP associated with chromatin, exhibiting a short residence time of 15 s (Fig. 4A–D). RNA interference-mediated depletion of the cohesin subunit Scc1 resulted in a small but significant reduction in the fraction of chromatin-bound Phf2-GFP (Fig. 4C; Appendix Fig. S4A) but did not alter the chromatin residence time of Phf2-GFP, suggesting that this is largely determined by interactions with H3K4me3-nucleosomes (Fig. 4D). Wapl depletion had the opposite effect, i.e., significantly increased the fraction of chromatin-bound Phf2 (Fig. 4C; Appendix Fig. S4A). Simultaneous depletion of both Wapl and Scc1 reversed this effect, suggesting that the enhanced chromatin association of Phf2 in Wapl-depleted cells is dependent on cohesin (Fig. 4C; Appendix Fig. S4A). As observed before (Kueng et al, 2006; Tedeschi et al, 2013), Wapl depletion also increased the fraction of cohesin bound to chromatin (Smc1-GFP, Appendix Fig. S4B–F). These results indicate that Wapl depletion increases the amount of chromatin-bound Phf2 by elevating the residence time and levels of cohesin on chromatin.

Conversely, Phf2 depletion did not detectably change the residence time or fraction of chromatin bound cohesin (Appendix Fig. S4B–F), suggesting that Phf2 is not required for the chromatin association of cohesin.

## Wapl and CTCF depletion re-positions Phf2 with cohesin in the genome

Our analysis of Phf2 in *Smc3* KO MEFs had indicated that cohesin contributes to the genomic distribution of Phf2 (Fig. 3). To test this

hypothesis further, we performed Phf2 ChIP-seq experiments using MEFs depleted of either Wapl or of Wapl and CTCF, in which cohesin accumulates in different regions than in wild type cells (Busslinger et al, 2017; Tedeschi et al, 2013). In *Wapl* KO MEFs, 5337 Phf2 peaks became undetectable at sites at which cohesin was also reduced, whereas 7313 new Phf2 peaks appeared at sites at which cohesin was enriched (Figs. 5A–C and EV3A–C), consistent with the co-accumulation of Phf2 and cohesin in vermicelli (Fig. 1B,C). Like cohesin (Haarhuis et al, 2017; Wutz et al, 2017), Phf2 became more enriched at CTCF sites (Fig. EV3C; see right panel) and at loop anchors detected by Hi-C (Fig. EV3D) following Wapl depletion. In contrast, Wapl depletion did not change the genomic distribution of H3K4me3, and no H3K4me3 could be detected at most of the Phf2 peaks newly formed in *Wapl* KO MEFs (Fig. EV3A–C,E,F). Consistent with these results, no accumulation of H3K4me3 in vermicelli was observed by immunofluorescence microscopy in Wapl depleted cells (Fig. EV3G). These results suggest that upon Wapl depletion, Phf2 and cohesin relocate together to new genomic loci.

In MEFs co-depleted of Wapl and CTCF, cohesin accumulates in regions that are several kb in size, compared to canonical cohesin peaks spanning ~0.5 kb. Most of these cohesin islands are located at sites of convergent transcription (Busslinger et al, 2017). In *Wapl-CTCF* KO MEFs, Phf2 accumulated in similar regions (Figs. 5D,E and EV4A,B) and, as observed for cohesin (Busslinger et al, 2017), the height and shape of these Phf2 islands correlated with the strength and symmetry with which the convergent gene pairs were transcribed (Figs. 5F and EV4C). These results support the notion that the distribution of Phf2 in the mouse genome is not only determined by H3K4me3-nucleosomes but also by cohesin. Previous work has shown that cohesin anchors new loops at cohesin islands, indicating that cohesin arrives in these locations by loop extrusion and is halted by transcription (Banigan et al, 2023). Our finding that Phf2 accumulates with cohesin in cohesin islands therefore suggests that Phf2 translocates with cohesin during loop extrusion (please note that Phf2's short residence time on chromatin is not inconsistent with this hypothesis; see Discussion).

## Phf2 limits the length of heterochromatic B compartments

Because Phf2 has been implicated in gene regulation and cell differentiation (Aguirre et al, 2024; Bricambert et al, 2018; Hata et al, 2013; Kim et al, 2014; Lane et al, 2019; Pappa et al, 2019; Shi et al, 2014; Stender et al, 2012), we tested whether Phf2 depletion causes gene expression defects in MEFs. However, only very few differences could be detected by RNA-seq between WT and *Phf2*

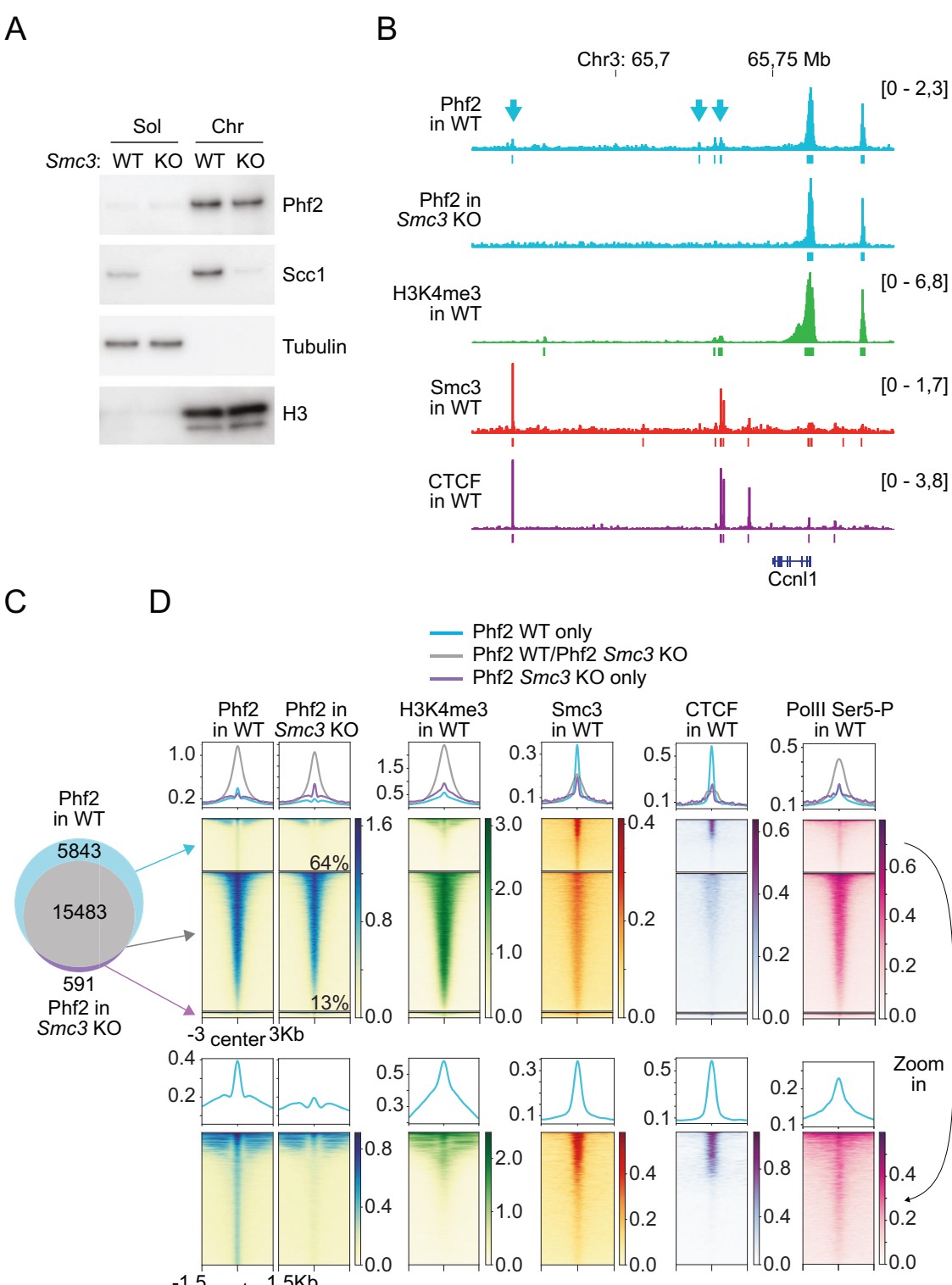

KO MEFs (Appendix Fig. S5A). These results suggest that Phf2 has only a minor, if any, gene regulatory role in quiescent MEFs, either because Phf2 is lacking such a function in quiescent cells, or because it functions redundantly with its paralogs Phf8 and Kdm7a.

Since Phf2 has been reported to de-methylate H3K9 in vitro (Baba et al, 2011; Bricambert et al, 2018; Horton et al, 2023), we tested whether Phf2 has a role in heterochromatin regulation. Hi-C experiments indeed showed a different distribution of euchromatic A and heterochromatic B compartments between WT and *Phf2* KO

**Figure 3.  Enrichment of Phf2 at sites lacking H3K4me3 is dependent on cohesin.**

(A) Immunoblot analysis of soluble (Sol) and chromatin-bound (Chr) fractions from WT and *Smc3* KO MEFs using the indicated antibodies. (B) Binding of Phf2 (in WT and *Smc3* KO MEFs), H3K4me3, Smc3, and CTCF at a representative locus, as determined by ChIP-seq. Small Phf2 peaks detectable in WT, but not in *Smc3* KO MEFs are indicated with blue arrows. (C) Venn diagram showing the overlap between ChIP-seq peaks of Phf2 in WT and Smc3 KO MEFs. (D) Pile-up heat maps and summary plots of ChIP-seq signals obtained for Phf2 (in WT and *Smc3* KO MEFs), H3K4me3, Smc3, CTCF, and PolII Ser5-P at the overlap groups indicated in (C). The Zoom-in panels show the indicated sub-groups of ChIP-seq signals at different color scales. Phf2 peaks present only in WT MEFs were reduced by 64% upon Smc3 depletion, whereas Phf2 peaks that were present in WT and *Smc3* KO MEFs were reduced by only 13%. Source data are available online for this figure.

MEFs (Fig. 6A). Eigenvector analyses revealed that B compartments were increased in length in Phf2 depleted cells, thus switched less frequently with A compartments, and correspondingly A compartments were reduced in length (Fig. 6C,E). As a result, the relative proportion of genomic sequences present in B compartments was increased, whereas those in A compartments was decreased (Fig. 6F,G). However, we could not detect changes in the distribution of active (H3K4me3) or inactive (H3K9me3) chromatin marks upon Phf2 depletion in ChIP-seq experiments (Appendix Fig. S5B–D). Phf2 therefore limits the length of B compartments, although it remains unclear whether Phf2 causes this effect as a histone demethylase or by other means.

Next we analyzed whether cohesin regulates Phf2's ability to limit B compartments. If so, one would predict that Wapl depletion has the opposite effect of Phf2 depletion, i.e., decreases the size of B compartments by allowing redistribution of Phf2 to genomic sites at which it is normally less abundant or absent. We therefore also analyzed compartments in *Wapl* KO MEFs. In Wapl depleted cells, compartmentalization is generally reduced (Haarhuis et al, 2017; Wutz et al, 2017), presumably as a result of increased cohesin-mediated loop extrusion (Nuebler et al, 2018; Schwarzer et al, 2017), but weak compartments can still be detected. Eigenvector analysis revealed that Wapl depletion had indeed the opposite effect of Phf2 depletion, i.e., shortened B compartments and increased A compartments (Fig. 6B,D,E). As a result, the proportion of genomic sequences contained in A compartments was increased and those in B compartments decreased (Fig. 6F,G). Similar observations have previously been made in human *WAPL* KO HAP1 cells where this phenomenon has been attributed to reduced levels of H3K9me3 (Haarhuis et al, 2022). However, in *Wapl* KO MEFs we could not detect such a reduction (Appendix Fig. S5B–D), consistent with our observation that no such changes could be detected in *Phf2* KO MEFs. It therefore remains unknown why A compartments increase at the expense of B compartments in *Wapl* KO MEFs, and why the opposite happens in *Phf2* KO MEFs.

### Phf2 contributes to the anchoring of short cohesin loops at H3K4me3-nucleosomes

We next analyzed whether Phf2 depletion affects genomic features other than compartments. The distribution of genomic distances (Appendix Fig. S6A) and most TAD boundaries were similar in *Phf2* KO MEFs and in WT cells (Appendix Fig. S6B). The insulation score was also not significantly altered in Phf2 depleted cells (Appendix Fig. S6C). However, a few TAD boundaries were not detected upon *Phf2* KO (472 boundaries of the 5,337 detected in WT), and these were characterized by lower levels of CTCF and Smc3 (Appendix Fig. S6D,E). These results suggest that Phf2 helps to establish a small subset of TADs, which are characterized by less CTCF binding at their boundaries.

To explore whether Phf2 affects loop formation we analyzed corner peaks in Hi-C data (Rao et al, 2014). This revealed that of 17,598 corner peaks present in WT cells, 4591 (26%) could not be detected in *Phf2* KO MEFs (Fig. 7A,B). On average, the loops lost in Phf2 depleted cells were significantly shorter (median 100 kb) than the loops that persisted after Phf2 depletion (median 270 kb; Fig. 7D). Interestingly, CTCF was present at both loop anchors in only 65% of the short Phf2-dependent loops, compared to 90% of the Phf2-independent loops (Fig. 7B). Moreover, we detected a significantly lower ChIP-seq signal for Smc3, and particularly for CTCF, at the anchors of Phf2-dependent loops compared to Phf2-independent loops (Fig. 7C). In contrast, the levels of H3K4me3 were higher at the anchors of Phf2-dependent loops (Fig. 7C), suggesting that many of these loops are anchored at H3K4me3-nucleosomes. These results indicate that Phf2 contributes to the accumulation of short cohesin loops at H3K4me3-nucleosomes at which CTCF occupancy is low.

### Phf2 contributes to the enrichment of cohesin at TSSs lacking CTCF

If Phf2 is required for the anchoring of cohesin loops at H3K4me3-nucleosomes, one would expect that cohesin also accumulates at these sites in a Phf2-dependent manner. Cohesin is in fact known to accumulate at active TSSs, which are flanked by H3K4me3-nucleosomes, independent of whether these TSSs contain a CTCF binding site or not (Schmidt et al, 2010). But how TSSs and possibly other sites containing H3K4me3-nucleosomes function as boundaries for loop extruding cohesin complexes is unknown. We therefore tested whether Phf2 is required for the accumulation of cohesin at these sites by analyzing the distribution of cohesin in *Phf2* KO MEFs.

ChIP-seq experiments revealed that 7243 small Smc3 peaks colocalizing with Phf2 could not be detected in *Phf2* KO MEFs (Fig. 8A red arrows, B, C), while 7954 larger Smc3 peaks that also overlapped with Phf2 persisted in these cells (Fig. 8A–C). The loss of small Smc3 peaks was not caused by a general reduction in cohesin levels since similar amounts of Smc3 were detected by immunoblotting in *Phf2* KO and WT cells, both in whole-cell extracts and on chromatin (Fig. 8D). Interestingly, the Phf2-dependent Smc3 peaks contained more Phf2 and H3K4me3 signal than Phf2-independent Smc3 peaks (Fig. 8E), suggesting that Phf2 bound H3K4me3-nucleosomes are required for the enrichment of cohesin at these sites.

Remarkably, most of the Smc3 sites that were sensitive to Phf2 depletion contained little if any CTCF, whereas most of the Smc3 peaks that remained after Phf2 depletion did overlap with CTCF (Figs. 8B,C,E and EV5A–C). These results suggest that cohesin can be recruited to TSSs either by CTCF or by Phf2. Supporting this hypothesis, pile-up analyses revealed that Smc3 ChIP-seq signals showed different distributions at TSSs, depending on whether these

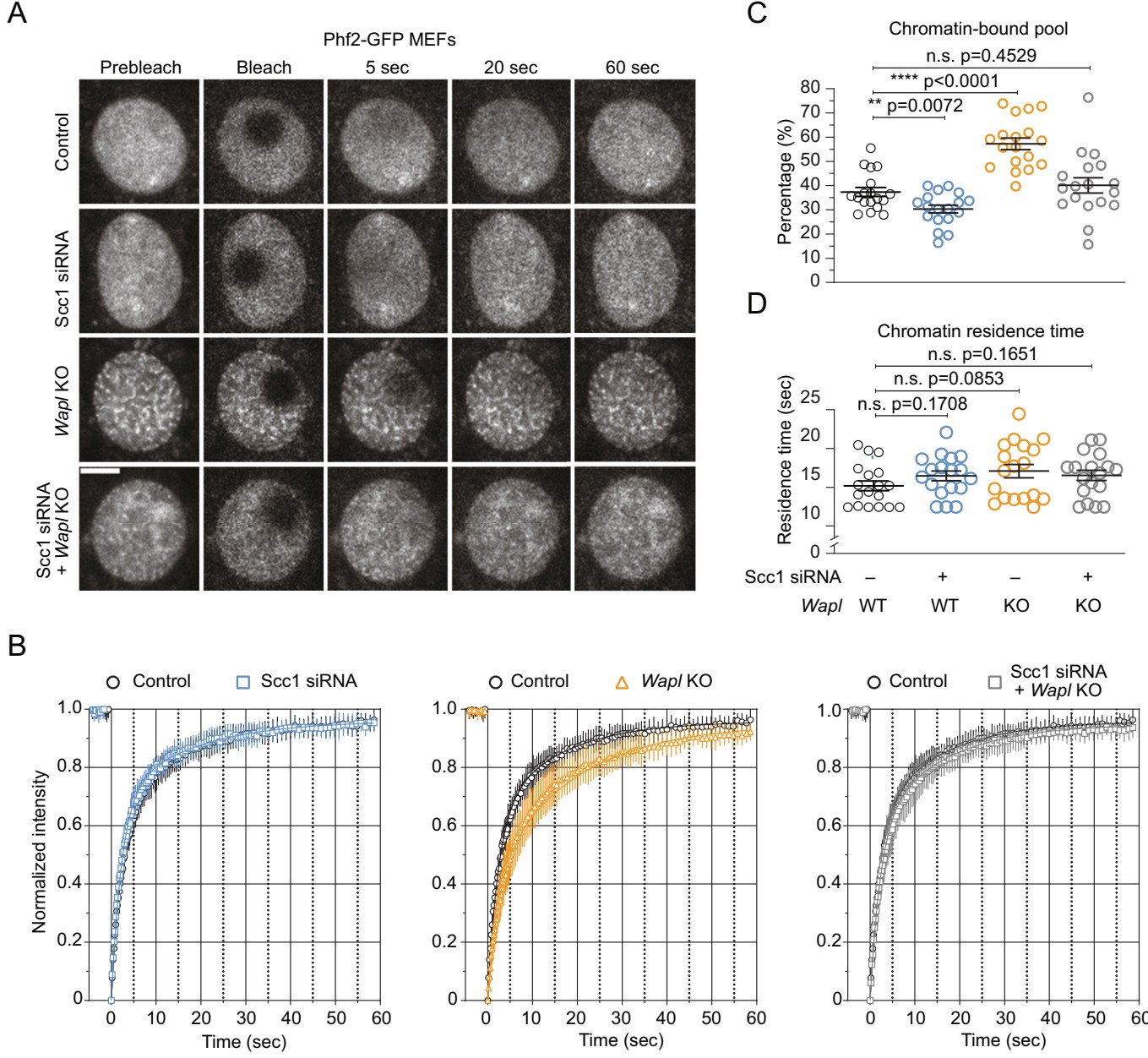

Figure 4. Cohesin influences Phf2 chromatin association.

(A) Microscopy images of MEFs expressing Phf2-GFP analyzed by fluorescence recovery after photobleaching (FRAP) of an area with 2 μm diameter. Scc1 was depleted by siRNA transfection in either control or *Wapl* KO MEFs. Scale bar, 2 μm. (B) Quantification of Phf2-GFP fluorescence intensity in the bleached area over time, shown for the conditions described in (A). Number of cells used n = 18. Error bars indicate the Standard Error of the Mean. (C) Quantification of the chromatin bound fraction of Phf2-GFP calculated from curves in (B). Statistical analyses were performed with unpaired t test and the *P* values are indicated in the figure. Error bars indicate the Standard Error of the Mean. (D) Quantification of the residence time on chromatin of Phf2-GFP calculated from curves in (B). Statistical analyses were performed with unpaired t test and the *P* values are indicated in the figure. Error bars indicate the Standard Error of the Mean. Source data are available online for this figure.

contained CTCF or not. In the presence of CTCF, Smc3 showed the highest signal enrichment at the CTCF peak but displayed a broader distribution towards the Phf2-bound +1-nucleosome (Fig. 8F). However, at TSSs lacking CTCF, Smc3 signals peaked at the +1-nucleosome position, where Phf2 and H3K4me3 are most enriched (Fig. 8F). These results indicate that both CTCF and Phf2 contribute to the enrichment of cohesin at TSSs, and that Phf2 becomes dominant in this role in the absence of CTCF.

## Discussion

### Phf2 is a cohesin interacting protein

It is well established that genome regulation depends on histone modifications and cohesin mediated loop extrusion. However, except for the proposed spreading of the DNA damage mark γH2AX by cohesin (Arnould et al, 2021) it is unknown whether

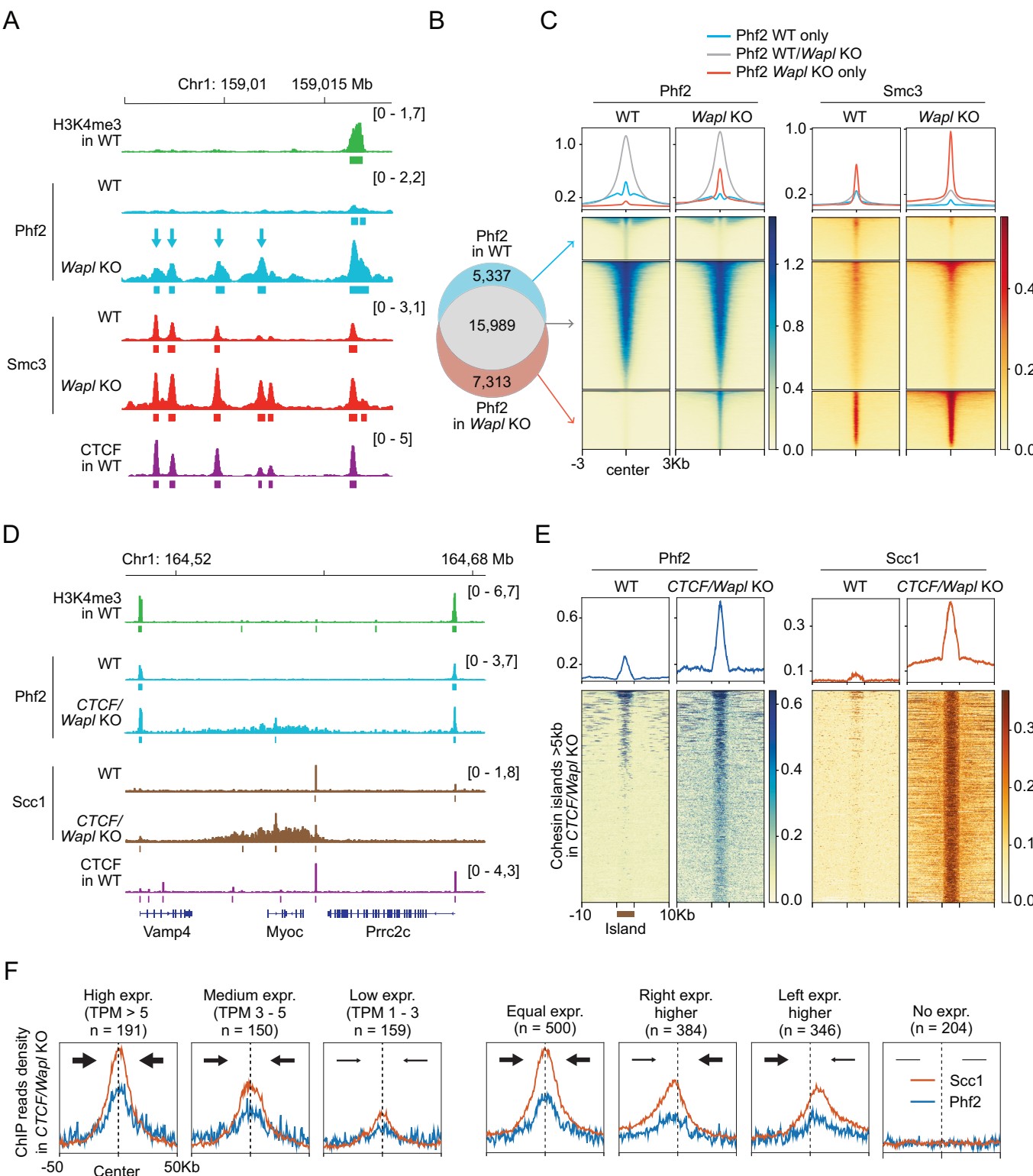

these mechanisms cooperate with each other. Our results provide important insight into this question by showing that the epigenetic reader and putative histone demethylase Phf2 is a cohesin interacting protein, suggesting a direct link between epigenetic and architectural mechanisms of genome regulation.

Our observations that Phf2 becomes enriched in cohesin immunoprecipitates following Wapl depletion, colocalizes with cohesin in vermicelli, and is present at thousands of genomic cohesin binding sites even in the presence of Wapl suggest that cohesin-Phf2 interactions occur frequently and at numerous sites in the genome.

**Figure 5. Wapl and CTCF depletion re-positions Phf2 with cohesin in the genome.**

(A) Binding of H3K4me3, Phf2 (in WT and *Wapl* KO MEFs), Smc3 (in WT and *Wapl* KO MEFs), and CTCF at a representative locus, as determined by ChIP-seq. Phf2 peaks detected only in *Wapl* KO MEFs are indicated with blue arrows. (B) Venn diagram showing the overlap between ChIP-seq peaks of Phf2 in WT and *Wapl* KO MEFs. (C) Pile-up heat maps and summary plots of ChIP-seq signals obtained for Phf2 (in WT and *Wapl* KO MEFs) and Smc3 (in WT and *Wapl* KO MEFs) at the overlap groups indicated in (B). (D) Binding of H3K4me3, Phf2 (in WT and *CTCT/Wapl* KO MEFs), Scc1 (in WT and *CTCT/Wapl* KO MEFs), and CTCF at a representative locus, as determined by ChIP-seq. (E) Pile-up heat maps and summary plots of ChIP-seq signals obtained for Phf2 (in WT and *CTCT/Wapl* KO MEFs) and Scc1 (in WT and *CTCT/ Wapl* KO MEFs) at cohesin islands found in *CTCT/Wapl* KO MEFs bigger than 5 kb (islands were stretched to equal size). (F) Summary plots of ChIP-seq signals obtained for Scc1 and Phf2 in *CTCT/Wapl* KO MEFs for cohesin islands at which convergent gene pairs are transcribed at similar levels: high (TPM > 5), medium (TPM 3–5), and low (TPM 1–3) (left 3 panels), or at which convergent gene pairs are transcribed symmetrically (Equal expr.) or asymmetrically (Right expr. higher; Left expr. higher) or with no detectable expression (No expr) (right 4 panels), as measured by RNA-seq in Busslinger et al, 2017. Source data are available online for this figure.

Our results further indicate that these interactions are direct, since purified Phf2 can bind to recombinant cohesin, and since a Phf2 mutant that is unable to bind cohesin in vitro is no longer enriched in vermicelli. Whereas we discovered Phf2 to be a cohesin interacting protein in MEFs, cohesin has recently also been identified as a Phf2 interacting protein in murine neural stem cells (NSCs) (Feng et al, 2024), further supporting the notion that these proteins are bona fide interactors.

We found that Phf2 binding depends on cohesin's STAG and kleisin subunits. Interestingly, αFold predicts that one of two Phf2's binding sites on this STAG-kleisin module is the same CES (conserved essential sequence) that CTCF binds to (Li et al, 2020). Mutation of a conserved YEY motif in Phf2 that is predicted to bind to this CES abolished the accumulation of Phf2 in vermicelli, suggesting that this motif and its interaction with the CES are indeed required for cohesin binding. It will therefore be interesting to test whether Phf2 and CTCF bind cohesin in a mutually exclusive manner and whether competition between these proteins serves a regulatory function. Our observation that Phf2 ChIP-seq peaks can overlap with CTCF peaks does not rule out this possibility since the resolution of these data is not high enough to determine whether Phf2 and CTCF are located at precisely the same genomic sites. In fact, our genome-wide pile-up analyses indicate that this is not the case and that Phf2 rather co-localizes with cohesin than with CTCF (Fig. EV1D). Furthermore, the ChIP-seq signals obtained in our experiments represent the distribution of Phf2, CTCF and cohesin in large cell populations and do not reveal whether these proteins are bound to the same genomic sites in individual cells.

## Phf2 might be translocated through the genome by cohesin-mediated loop extrusion

Phf2 is an epigenetic reader that binds H3K4me3-nucleosomes in the vicinity of active TSSs (Bricambert et al, 2018; Pappa et al, 2019). However, our finding that Phf2 is also detected at many cohesin sites lacking H3K4me3, and that cohesin depletion reduces Phf2 enrichment at these sites, suggests that the genomic positioning of Phf2 is determined not only by H3K4me3-nucleosomes but also by cohesin.

Since Phf2 binds cohesin we propose that Phf2 translocates through the genome by cohesin-mediated loop extrusion. Consistent with this possibility, Wapl depletion results in a dramatic re-localization of Phf2 into vermicelli, in which cohesin is thought to accumulate by loop extrusion (Fudenberg et al, 2016; Haarhuis

et al, 2017; Wutz et al, 2017). Similarly, Phf2 accumulates together with cohesin at sites of convergent transcription in cells depleted of Wapl and CTCF. These cohesin islands form new chromatin loops (Banigan et al, 2023), implying that cohesin—and thus Phf2 bound to it—translocate to these sites by loop extrusion. Our finding that Phf2 has a chromatin residence time (~15 s) much shorter than that of cohesin (~5 min; (Gerlich et al, 2006)) does not contradict this hypothesis, as NIPBL also has a short chromatin residence time (~50 s; (Rhodes et al, 2017)) but translocates with cohesin during loop extrusion and is in fact essential for this process (Barth et al, 2023; Davidson et al, 2019).

## Phf2 limits the size of B compartments

The hypothesis that Phf2 'travels' with cohesin during loop extrusion raises the intriguing possibility that cohesin might spatially control epigenetic functions that Phf2 might have. These functions are poorly understood, possibly because Phf2 acts redundantly with paralogous histone de-methylases such as Phf8 and Kdm7a, or because Phf2 does not function as a histone de-methylase in cells but regulates chromatin through unknown other mechanisms. The latter possibility is consistent with the presence of an unusual histidine-to-tyrosine exchange in Phf2's JmjC domain (Horton et al, 2011). Phf2 shares this feature with fission yeast Epe1, which regulates the genomic distribution of H3K9me3 in vivo but has no detectable histone demethylase activity in vitro (Ayoub et al, 2003; Braun et al, 2011; Ragunathan et al, 2015; Trewick et al, 2007).

In either case, our finding that Phf2 depletion increases the size of B compartments indicates that Phf2 limits the size of heterochromatin domains, as does Epe1 in fission yeast. Interestingly, the opposite happens in Wapl depleted cells. In these cells, B compartments decrease in size ((Haarhuis et al, 2022); this study) as one would predict if prolonged cohesin mediated loop extrusion would translocate Phf2 into heterochromatic regions in which Phf2 is normally less enriched. These results therefore suggest that Phf2 limits the size of B-compartments in a manner that is spatially regulated by cohesin-mediated loop extrusion.

## Phf2 recruits cohesin to active TSSs that lack a CTCF binding site

Although CTCF sites are the most frequent boundary elements in mammalian genomes at which cohesin complexes accumulate (Parelho et al, 2008; Wendt et al, 2008), cohesin is also recruited to

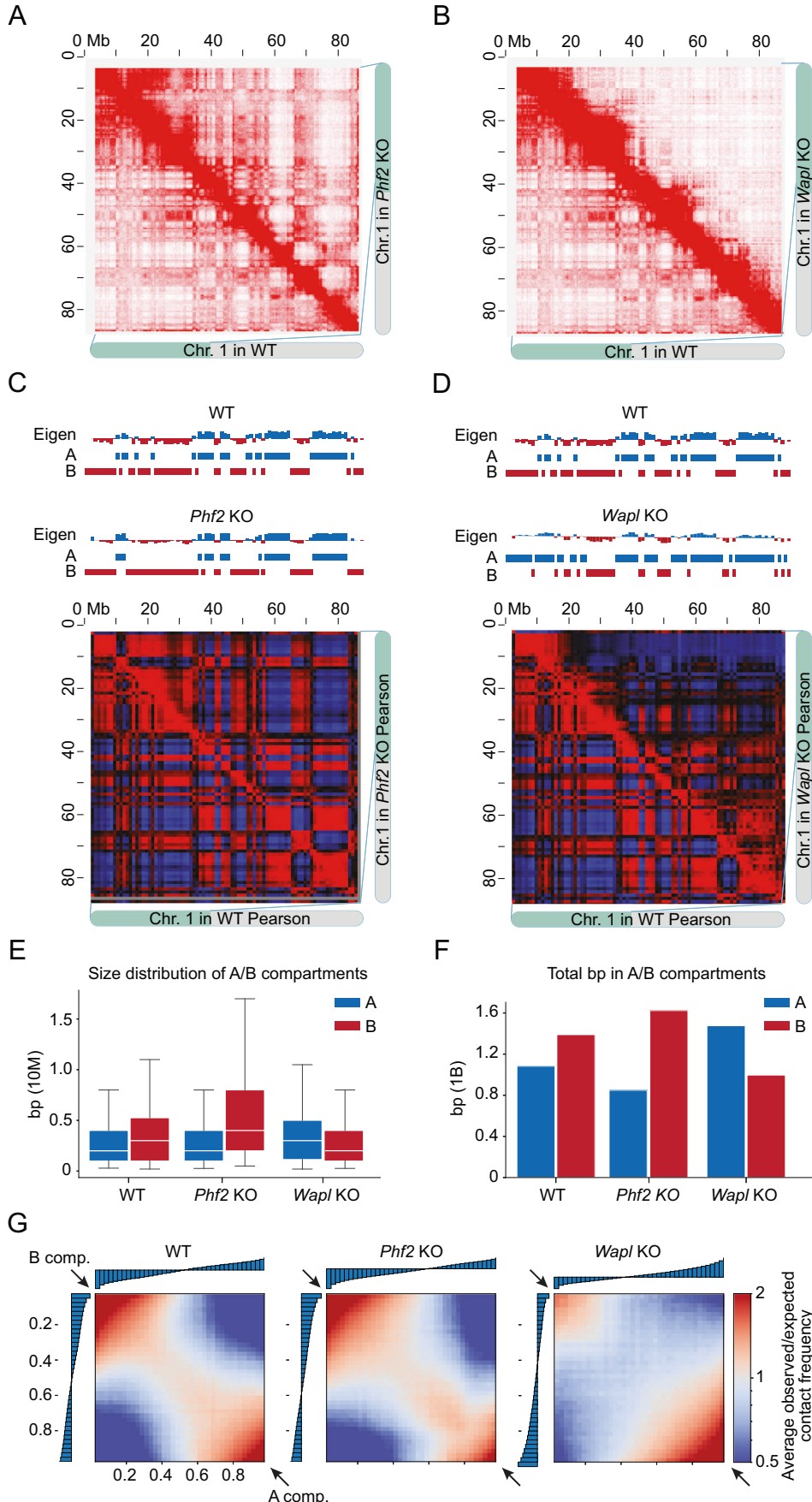

**Figure 6.   Phf2 depletion limits the length of heterochromatic B compartments.**

(A) Chromosome compartmentalization was visualized using raw contact data from coverage-corrected Hi-C contact matrices of chromosome 1 (1–85 Mb). WT data are shown in the lower left half, while data from *Phf2* KO MEFs are shown in the top right half. (B) Contact matrices as in (A), obtained from WT (lower left half) and *Wapl* KO MEFs (top right half). (C) Distribution of active (A) and inactive (B) compartments shown by Eigenvector analysis (top) and as a Pearson autocorrelation matrix (bottom). Data from WT are shown in the lower left half of the matrix, data from *Phf2* KO MEFs in the top right half. Positive values corresponding to A compartments are shown in blue, and negative values corresponding to B compartments in red. (D) Compartment distribution as in (C), obtained from WT (lower left half) and *Wapl* KO MEFs (top right half). (E) A box and whisker plot illustrating the size distribution of A and B compartments in WT, *Phf2* KO, and *Wapl* KO MEFs. The box around the median indicates the interquartile range (IQR), while the whiskers indicate 1.5 IQR. Biological replicates $n = 2$. (F) A column plot illustrating the fraction of the genome that is present in A and B compartments in WT, *Phf2* KO, and *Wapl* KO MEFs. (G) Saddle plots illustrating the distribution of A and B compartments in WT, *Phf2* KO, and *Wapl* KO MEFs. Source data are available online for this figure.

active TSSs that do not contain CTCF binding sites (Schmidt et al, 2010). The molecular basis for this CTCF-independent accumulation of cohesin at TSSs was unknown, but our results indicate that Phf2 is required for this effect. Since Phf2 can bind to H3K4me3-nucleosomes and cohesin via distinct domains, a N-terminal PHD finger and a C-terminal cohesin binding domain, Phf2 might recruit cohesin to TSSs by simultaneously binding to H3K4me3-nucleosomes and cohesin. Interestingly, these results suggest that all three proteins that are known to function as boundaries for loop extruding cohesin interact with cohesin's CES (conserved essential sequence) via similar motifs, an N-terminal YxY motif in case of CTCF (Li et al, 2020), a YDF motif in MCM's MCM3 subunit (Dequeker et al, 2022; Li et al, 2020), and a YEY motif in Phf2's C-terminus.

Our results indicate that cohesin accumulates in distinct regions at TSSs depending on whether these contain a CTCF binding site or not. In the presence of CTCF, most cohesin signal is detected at the CTCF site, but at TSSs lacking a CTCF site the peak of cohesin is detected at the position of the +1-nucleosome at which Phf2 accumulates. These results suggest that cohesin can be recruited to TSSs through two distinct mechanisms: interactions with CTCF and Phf2-mediated interactions with H3K4me3-nucleosomes. We propose that both of these interactions contribute to the ability of active genes to function as boundaries for loop extruding cohesin complexes (Banigan et al, 2023). Consistent with this possibility we find that Phf2 depletion abrogates short cohesin loops lacking CTCF, many of which are anchored at TSSs.

## Cooperation between epigenetic and architectural mechanisms in genome regulation

Our results reveal unexpected interactions between epigenetic and architectural mechanisms of genome regulation by showing that loop extruding cohesin complexes interact with the epigenetic reader protein Phf2 and contribute to its genomic positioning. If cohesin complexes perform this function by translocating Phf2 via loop extrusion, as our results suggest, this interaction could be conceptually similar to how cytoskeletal motor proteins translocate cargo molecules along microtubules (Vale, 2003). It will therefore be interesting to test whether cohesin complexes can also translocate other molecules than Phf2 by loop extrusion, for example to facilitate the spreading of histone marks (Arnould et al, 2021) or to recruit proteins to TAD boundaries and active TSSs. For Phf2 itself, it will be important to address in future experiments how it limits the size of B compartments and whether it functions as a de-methylase for histones or possibly for cohesin itself.

# Methods

**Reagents and tools table**

| Reagent/Resource | Reference or Source | Identifier or Catalog Number |
|---|---|---|
| **Experimental models** | | |
| C57BL/6J (*M. musculus*) | Jackson Lab | |
| MEFs (*M. musculus*) | This study | |
| **Recombinant DNA** | | |
| MigR1 | Addgene | #27490 |
| pLIB | Addgene | #80610 |
| MigR1-Phf2 | This Study | |
| MigR1-ΔPHD | This Study | |
| MigR1-ΔJmjC | This Study | |
| MigR1-Phf2 Δ1 | This Study | |
| MigR1-Phf2 Δ2 | This Study | |
| MigR1-Phf2 Δ3 | This Study | |
| MigR1-Phf2 Δ4 | This Study | |
| MigR1-Phf2 Y673A Y675A | This Study | |
| MigR1-Phf2 Y869A Y871A | This Study | |
| MigR1-Phf2 Δ673–711 | This Study | |
| pLIB-His-Phf2 | This Study | |
| pLIB-His-Phf2 Δ1 | This Study | |
| pLIB-His-Phf2 Δ2 | This Study | |
| pLIB-His-Phf2 Δ3 | This Study | |
| pLIB-His-Phf2 Δ4 | This Study | |
| **Antibodies** | | |
| Rabbit anti-Phf2 | Cell Signaling | 3497S |
| Rabbit anti-Smc3 | Peters laboratory | ID A941 |
| Mouse anti-Scc1 | Upstate | 05-908 |
| Rabbit anti-Sa1 | Peters laboratory | ID A823 |
| Rabbit anti-Sa2 | Peters laboratory | ID A824 |
| Rabbit anti-Wapl | Peters laboratory | ID A960 |
| Rabbit anti-CTCF | Peters laboratory | ID A992 |
| Rabbit anti-MCM3 | Bethyl | A300-192A |

| Reagent/Resource | Reference or Source | Identifier or Catalog Number |
| --- | --- | --- |
| Mouse anti-MCM5 | Santa Cruz | sc-136366 |
| Mouse anti-Tubulin | Sigma | T5168 |
| Goat anti-H3 | Santa Cruz | sc-8654 |
| Mouse anti-GFP | Roche | 11814460001 |
| Rabbit anti-Scc1 | Peters laboratory | ID A890 |
| Rabbit anti-Brwd1 | Sigma | HPA030945 |
| Rabbit anti-Top2b | Peters laboratory | ID A1026 |
| Rabbit anti-Chd4 | Active Motif | 39290 |
| Rabbit anti-Baz1b | Millipore | MABE194 |
| Rabbit anti-Phf8 | Abcam | Ab36068 |
| Mouse anti-FLAG | Sigma | F1804 |
| Chicken anti-GFP | Abcam | Ab13970 |
| Rabbit anti-H3K4me2 | Millipore | 07-030 |
| Rabbit anti-H3K4me3 | Abcam | Ab8580 |
| Rabbit anti-H4K20me3 | Abcam | Ab9053 |
| Rabbit anti-H3K9me1/2/3 | Jenuwein laboratory | No. 4858/4677/4861 |
| Rabbit anti-Smc3 | Bethyl | A300-060A |
| Rabbit anti-CTCF | Upstate | 07-729 |
| Mouse anti-RNA polymerase II S5 | Abcam | Ab5408 |
| **Oligonucleotides and other sequence-based reagents** | | |
| Phf2 siRNA | Dharmacon | L-058476-01-0020 |
| Scc1 siRNA | Ambion | ID s72658 |
| **Chemicals, Enzymes and other reagents** | | |
| 4-hydroxytamoxifen | Sigma | H7904 |
| Benzonase | Millipore | #70664 |
| **Software** | | |
| Samtools 1.10 | https://www.htslib.org/ | |
| Bamtools 2.5.1 | https://github.com/pezmaster31/bamtools | |
| Bedtools 2.27.1 | https://bedtools.readthedocs.io/en/latest/ | |
| Igvtools 2.4.18 | https://igv.org/doc/desktop/ | |
| Cooler 0.8.8 | https://cooler.readthedocs.io/en/latest/index.html | |
| hic2cool 0.8.3 | https://github.com/4dn-dcic/hic2cool | |
| Python 3.6.6 | https://www.python.org/ | |
| Pandas 0.25.3 | https://pandas.pydata.org/ | |
| macs2 2.2.5 | https://github.com/macs3-project/MACS | |
| Java 1.8.0_212 | https://www.java.com/en/ | |
| Open2c | https://open2c.github.io/ | |
| GraphPad Prism 9 | https://www.graphpad.com | |

| Reagent/Resource | Reference or Source | Identifier or Catalog Number |
| --- | --- | --- |
| ImageJ | https://imagej.nih.gov/ij/index.html | |
| **Other** | | |

## Antibodies

The following antibodies were used for immunoblot analysis: rabbit anti-Phf2 (Cell Signaling, 3497S), rabbit anti-Smc3 (Peters laboratory ID A941), mouse anti-Scc1 (Upstate 05-908), rabbit anti-Sa1 (Peters laboratory ID A823), rabbit anti-Sa2 (Peters laboratory ID A824), rabbit anti-Wapl (Peters laboratory ID A960), rabbit anti-CTCF (Peters laboratory ID A992), rabbit anti-MCM3 (Bethyl A300-192A), mouse anti-MCM5 (Santa Cruz sc-136366), mouse anti-Tubulin (Sigma T5168), goat anti-H3 (Santa Cruz sc-8654) and mouse anti-GFP (Roche, 11814460001).

The following antibodies were used for immunofluorescence microscopy: rabbit anti-Phf2 (Cell Signaling, 3497S), mouse anti-Scc1 (Upstate, 05-908) and rabbit anti-Scc1 (Peters laboratory, ID A890), rabbit anti-Brwd1 (Sigma, HPA030945), rabbit anti-Top2b (Peters laboratory ID A1026), rabbit anti-Chd4 (Active Motif, 39290), rabbit anti-Baz1b (Millipore, MABE194), rabbit anti-Phf8 (Abcam, Ab36068), mouse anti-FLAG (Sigma, F1804), chicken anti-GFP (Abcam, Ab13970), rabbit anti-H3K4me2 (Millipore 07-030), rabbit anti-H3K4me3 (Abcam, Ab8580), rabbit anti-H4K20me3 (Abcam, Ab9053), and rabbit anti-H3K9me1/2/3 (No. 4858/4677/4861, gifts from Thomas Jenuwein).

The following antibodies were used for ChIP experiments: rabbit anti-Phf2 (Cell Signaling, 3497S), rabbit anti-H3K4me3 (Abcam, Ab8580), rabbit anti-H3K9me3 (Abcam, Ab8898), rabbit anti-Smc3 (Bethyl A300-060A), rabbit anti-CTCF (Upstate 07-729), and mouse anti-RNA polymerase II S5 (Abcam, Ab5408).

## DNA constructs

Phf2/Phf8/Smc1-LAP BACs (the 'localization and affinity purification' [LAP] tag consists of GFP for localization and immunopurification; hereafter referred to as Phf2/Phf8/Smc1-GFP) were gifts from Anthony Hyman (MPI-CBG, Dresden, Germany) and were introduced into cells using Fugene HD transfection reagents, followed by FACS sorting based on GFP expression levels. The full-length Phf2 and truncated mutants were amplified using the DNA template P3FLAG-Phf2 (gift from Xiaobing Shi, The University of Texas MD Anderson Cancer Center, Houston, TX), and PCR products were subcloned into the MigR1 or pLIB vector.

## Cell culture and activation of Cre

Primary MEFs (pMEFs) were generated from E13.5 embryos and cultured in full medium (DMEM supplemented with 10% FBS, 0.2 mM L-glutamine, 100 U/ml penicillin, 100 µg/ml streptomycin, 1 mM sodium pyruvate, 0.1 mM 2-mercaptoethanol, and non-essential amino acids). Immortalized MEFs (iMEFs) were cultured in the same medium as pMEFs. Phoenix cells were cultured in DMEM

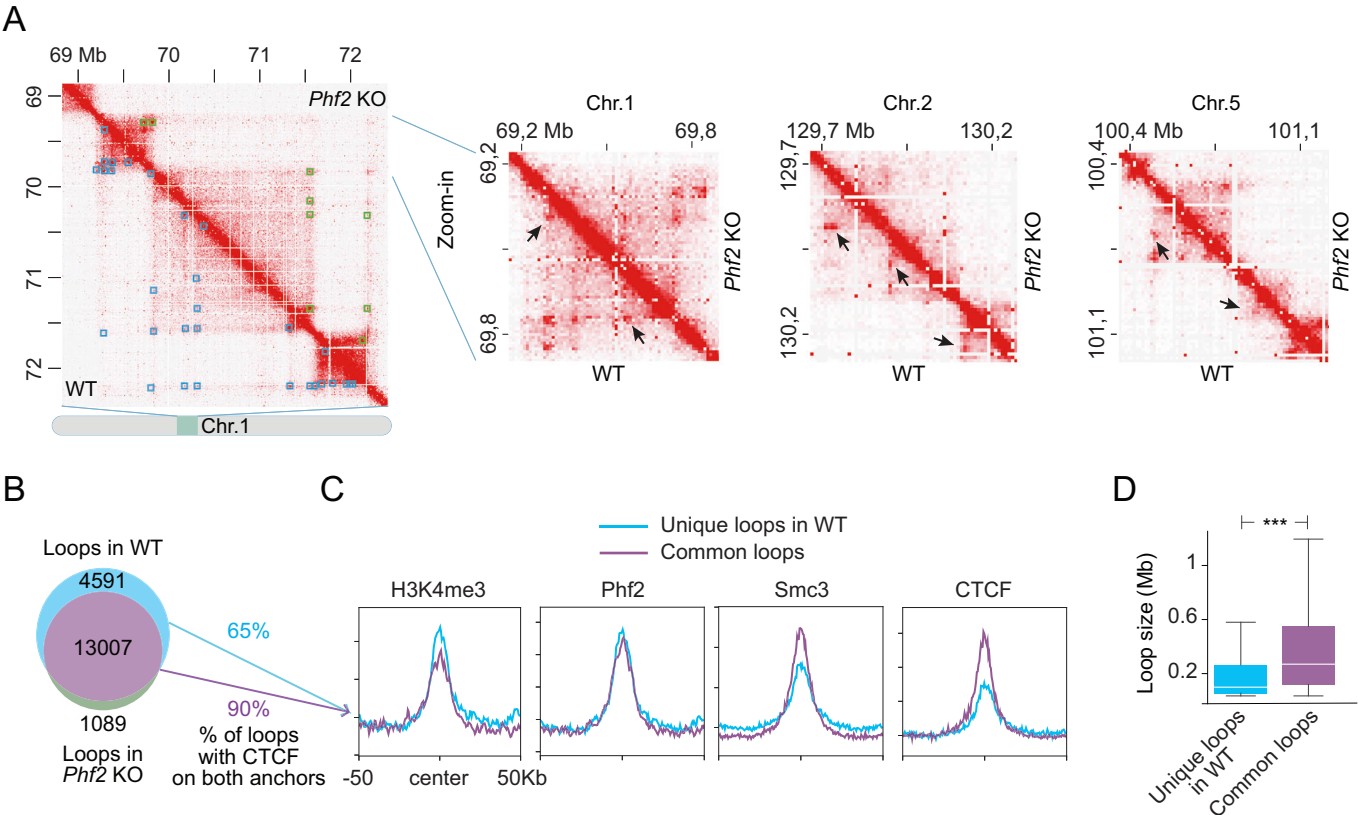

**Figure 7. Phf2 depletion reduces a subset of short chromatin loops with less CTCF at their anchor sites.**

(A) Chromosome contacts were visualized using raw contact data from coverage-corrected Hi-C matrices of chromosome 1 (69–72 Mb). Data from WT MEFs are shown in the lower left half, data from *Phf2* KO MEFs in the top right half. Loops detected by Juicer tool in WT and *Phf2* KO MEFs are depicted with blue (bottom) and green (top) squares, respectively. Loops that were sensitive to Phf2 depletion are indicated with black arrows on matrixes from three representative chromosomal regions (right 3 panels). (B) Venn diagram showing the overlap between loops in WT and *Phf2* KO MEFs. Numbers indicate the percentage of loops with CTCF peaks detected at both anchor sites. (C) Summary plots of ChIP-seq signals obtained for H3K4me3, Phf2, Smc3, and CTCF at the indicated loop anchors. (D) Box and whisker plot depicting the sizes of loops that are detectable in WT and *Phf2* KO MEFs or only in WT MEFs. The box around the median indicates the interquartile range (IQR), while the whiskers indicate 1.5 IQR. Statistical analyses were performed with t test and the P values $= 3.75^{E-09}$. Biological replicates $n = 2$. Source data are available online for this figure.

supplemented with 10% FBS, 0.2 mM L-glutamine, 100 U/ml penicillin, and 100 µg/ml streptomycin.

All experiments with MEFs were performed in quiescence. To achieve this, MEFs were grown to confluence, and then the full medium was replaced with resting medium. For the conditional deletion of floxed Phf2, Wapl, and Smc3 alleles, we used the inducible CreERT2 recombinase expressed from the Rosa26 locus (CreER). Cells were arrested in Opti-MEM (Invitrogen) supplemented with 2% charcoal-treated FBS, 100 U/ml penicillin, 100 µg/ml streptomycin, and 0.5 µM 4-hydroxytamoxifen (4-OHT, Sigma) for 7 days. For the conditional deletion of the floxed CTCF allele, MEFs were cultured for 4 h in resting medium (DMEM supplemented with 2% FBS), 4 µg/ml polybrene, and Adeno-Cre virus (Penn Vector Core, 1 µl per 1 million cells). The medium was then replaced with resting medium for 7 days.

## Chromatin immunoprecipitation (ChIP)–mass spectrometry (MS)

Cells cultured in trays (Thermo, cat#166508) were cross-linked, washed with cold PBS, scraped off, and snap-frozen. The cell pellets

were resuspended in extraction buffer (25 mM Tris, pH 7.5, 100 mM NaCl, 5 mM MgCl₂, 0.2% NP-40, 10% glycerol, EDTA-free protease inhibitor tablet, and 1 mM PMSF). After brief sonication (2 × 10 s) and centrifugation (1700 × g, 2 min), the pellets were resuspended in extraction buffer supplemented with 90 U/ml benzonase (Millipore, cat#70664) and rotated at 4 °C for 1 h. Ten volumes of RIPA buffer I (50 mM Tris, pH 7.5, 150 mM NaCl, 1 mM EDTA, 1% NP-40, 0.5% Na-deoxycholate, 0.1% SDS, and 1 mM PMSF) were added, followed by sonication (6 × 10 s). The supernatants were clarified by centrifugation at 13,000 rpm for 10 min and rotated overnight at 4 °C with 100 µl antibody-coupled beads (Herzog and Peters, 2005). Beads were then washed twice with RIPA buffer I, twice with RIPA buffer II (50 mM Tris, pH 7.5, 500 mM NaCl, 1 mM EDTA, 1% NP-40, 0.5% Na-deoxycholate, 0.1% SDS, and 1 mM PMSF), twice with RIPA buffer III (50 mM Tris, pH 7.5, 150 mM NaCl, 1 mM EDTA, and 1 mM PMSF), and twice with 150 mM NaCl. Bound protein complexes were eluted twice with 50 µl of 0.1 M glycine (pH 2.0), pooled, neutralized with 7 µl of 1.5 M Tris-HCl (pH 9.2), and used for silver staining, western blotting, or MS.

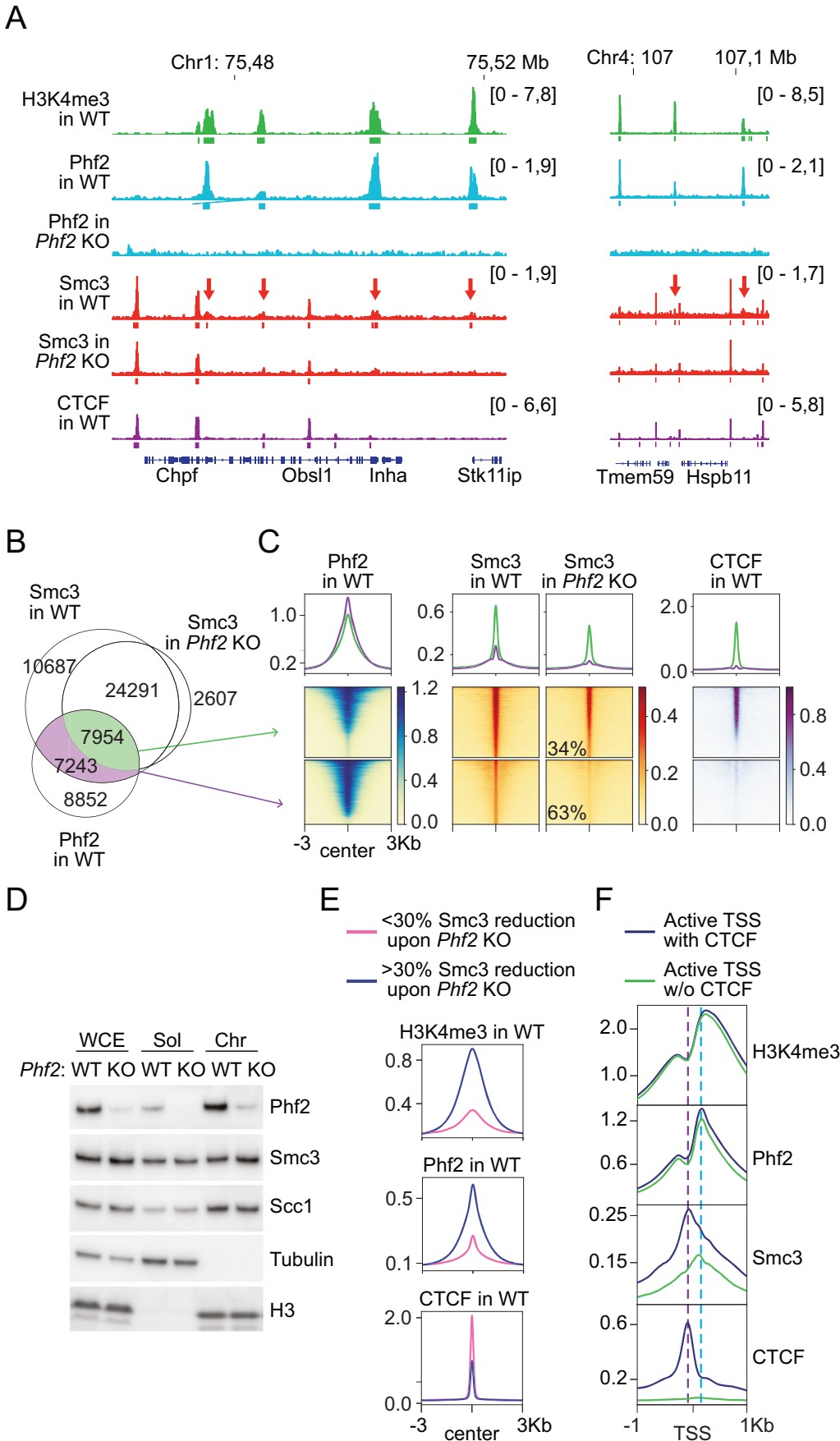

◀ **Figure 8. Phf2 localizes cohesin to TSSs in the absence of CTCF.**

(A) Binding of H3K4me3, Phf2 (in WT and *Phf2* KO MEFs), Smc3 (in WT and *Phf2* KO MEFs), and CTCF at two representative loci, as determined by ChIP-seq. Smc3 peaks detected only in WT, but not in *Phf2* KO MEFs are indicated with red arrows. (B) Venn diagram showing the overlap between ChIP-seq peaks obtained for Smc3 in WT and *Phf2* KO MEFs, with Phf2 peaks in WT MEFs. (C) Pile-up summary plots (top) and heat maps (bottom) of ChIP-seq signals obtained for Phf2 in WT MEFs, Smc3 (in WT and *Phf2* KO MEFs), and CTCF at the overlap groups indicated in (B). Numbers indicate the reduction of ChIP-seq signal of Smc3 peaks in *Phf2* KO MEFs. (D) Immunoblot analysis of whole-cell extract (WCE), soluble (Sol), and chromatin-bound (Chr) fractions from WT and *Phf2* KO MEFs using the indicated antibodies. (E) Summary plots of ChIP-seq signals for H3K4me3, Phf2, and CTCF at Smc3 peaks showing a reduction of more than or less than 30% upon Phf2 depletion. (F) Summary plots of ChIP-seq signals obtained for H3K4me3, Phf2, Smc3, and CTCF at active TSSs that contain CTCF or not. The purple dotted line indicates the apex of the cumulative CTCF signal, while the blue dotted line indicates the peak of the Phf2 signal. Source data are available online for this figure.

## NanoLC-MS analysis

The nano HPLC system used was an UltiMate 3000 RSLC nano system (Thermo Fisher Scientific, Bremen, Germany) coupled to an LTQ Orbitrap Velos mass spectrometer (Thermo Fisher Scientific, Bremen, Germany), equipped with a Proxeon nanospray source (Proxeon, Odense, Denmark). Peptides were loaded onto a trap column (Thermo Fisher Scientific, Bremen, Germany, PepMap C18, 5 mm × 300 μm ID, 5 μm particles, 100 Å pore size) at a flow rate of 25 μL min$^{-1}$ using 0.1% TFA as mobile phase. After 10 min, the trap column was switched in line with the analytical column (Thermo Fisher Scientific, Bremen, Germany, PepMap C18, 500 mm × 75 μm ID, 3 μm, 100 Å). Peptides were eluted using a flow rate of 230 nl min$^{-1}$, and a 3 h gradient, respectively, 180 min.

The gradient starts with the mobile phases: 98% A (water/formic acid, 99.9/0.1, v/v) and 2%B (water/acetonitrile/formic acid, 19.92/80/0.08, v/v/v) increases to 35%B over the next 180 min, followed by a gradient in 5 min to 90%B, stays there for five min and decreases in 5 min back to the gradient 98%A and 2%B for equilibration at 30 °C.

The LTQ Orbitrap Velos was operated in data-dependent mode, using a full scan in the Orbitrap (*m/z* range 350–2000, nominal resolution of 60,000, target value 1E6) followed by MS/MS scans of the 12 most abundant ions in the linear ion trap. MS/MS spectra (normalized collision energy 35%; activation value q 0.25; activation time 10 ms; isolation width 2, target value 1E4) were acquired and subsequent activation was performed on fragment ions through multistage activation. The neutral loss mass list was therefore set to −98, −49, and −32.6 *m/z*. Precursor ions selected for fragmentation (charge state 2 and higher) were put on a dynamic exclusion list for 60 s. Additionally, singly-charged parent ions were excluded from selection for MS/MS experiments and the monoisotopic precursor selection feature was enabled.

## MS data processing

For peptide identification, the RAW-files were loaded into Proteome Discoverer (version 2.1.0.81, Thermo Scientific). All MS/MS spectra were searched using MSAmanda 2.1.5.7869 (Dorfer et al, 2014). The peptide mass tolerance was set to ±5 ppm and the fragment mass tolerance to 0.03 Da, the maximal number of missed cleavages was set to 2, using tryptic enzymatic specificity without proline restriction. Peptide and protein identification was performed in two steps. For the initial search the RAW-files were searched against the database

swissprot_mouse.fasta, using the following search parameters: beta-methylthiolation of cysteine was set as a fixed modification, oxidation of methionine and deamidation of glutamine and asparagine as variable modifications. The result was filtered to 1% FDR on protein using the Percolator algorithm (Kall et al, 2007) integrated in Proteome Discoverer. A sub-database of proteins identified in this search was generated for further processing. For the second search, the RAW-files were searched against the created sub-database using the same settings as above plus considering additional variable modifications: Phosphorylation on serine, threonine, and tyrosine, acetylation on lysine, methylation and demethylation on lysine and arginine, trimethylation on lysine, ubiquitination on lysine. The localization of the post-translational modification sites within the peptides was performed with the tool ptmRS (v1.4.5735.14920) based on the tool phosphoRS (Taus et al, 2011). Identifications were filtered again to 1% FDR on protein and PSM level. Additionally, an MSAmanda score cut-off of at least 150 was applied. Peptides were subjected to label-free quantification using IMP-apQuant Version 2.18.83.16356 (Doblmann et al, 2019). Proteins were quantified by summing unique and razor peptides and applying intensity-based absolute quantification (iBAQ, (Schwanhausser et al, 2011)). Proteins were filtered to be identified by a minimum of 2 quantified peptides. Protein-abundances-normalization was done using sum normalization. Statistical significance of differentially expressed proteins was determined using limma (Smyth, 2004).

## Retrovirus production and infection

For retrovirus production, Phoenix cells were transiently transfected with MigR1-Phf2 or Phf2 mutants using the calcium phosphate transfection protocol, and viral supernatants were harvested 48 h post-transfection. For infection, MEFs were cultured in the presence of the viral supernatant and 8 μg/mL polybrene for 48 h.

## Chromatin fractionation

Cells were resuspended in extraction buffer (25 mM Tris, pH 7.5, 100 mM NaCl, 5 mM MgCl$_2$, 0.2% NP-40, 10% glycerol) supplemented with an EDTA-free protease inhibitor tablet (Roche) and 1 mM PMSF. The soluble fraction was separated from the chromatin by centrifugation. The chromatin pellet was washed three times with extraction buffer and digested with benzonase (Millipore) on ice for 10 min. For IP, centrifuge the lysate at 13,000 rpm for 10 min at 4 °C, and use the supernatant. For western blot, add loading buffer directly to the lysate.

## Silver staining of polyacrylamide gels

The gel was fixed with 10% (v/v) acetic acid and 45% (v/v) methanol for 15 min. The fixing solution was replaced with Farmer's Reducer (30 mM $K_3Fe(CN)_6$, 30 mM $Na_2S_2O_3$) for 2 min. The gel was then washed several times with MonoQ water until completely destained and incubated with 0.1% (w/v) silver nitrate for 15 min. After two brief washes with MonoQ water, the gel was incubated for 30 s with 2.5% (w/v) $Na_2CO_3$. The staining was developed by replacing the solution with freshly prepared 0.1% (v/v) formaldehyde in 2.5% $Na_2CO_3$. The reaction was stopped by adding 10% (v/v) acetic acid.

## Immunofluorescence

Cells were grown on coverslips, washed once with PBS, and then fixed for 20 min with 4% paraformaldehyde solution. The cells were permeabilized with 0.1% Triton X-100 in PBS for 5 min and blocked with 3% BSA in PBST (0.01% Triton X-100 in PBS) for 30 min before incubation with the primary antibody for at least 1 h. After washing with PBST, the secondary antibody was applied for 1 h, and nuclei were stained with DAPI for 5 min. The coverslips were mounted with ProLong Gold Antifade Reagent (Invitrogen) on microscope slides and analyzed by microscopy.

## Live cell imaging

iMEFs (Wapl F/F, ERCre/+) expressing Phf2-GFP were seeded on Lab-Tek chambered coverslips (Thermo Scientific Nunc) and treated with or without 4-OHT. Before imaging, the culture medium was replaced with pre-warmed $CO_2$-independent medium. DNA was visualized by staining with Hoechst. Imaging was performed using an LSM780 confocal microscope (Carl Zeiss) with a 63×/1.4 numerical aperture (N/A) objective.

## Fluorescence recovery after photobleaching (FRAP)

FRAP was performed as described (Ladurner et al, 2014). iMEFs (Wapl F/F, ERCre/+) expressing Phf2/Smc1-GFP were imaged on a Zeiss LSM5 Duo confocal microscope using a 63× Plan-Apochromat objective and an open pinhole. Twenty images were acquired before bleaching a radial spot ($r = 2$ μm) three times at 100% laser intensity (100 mW diode 488). Bleaching with high laser power allowed for a short total bleach time, keeping intra-bleach diffusion effects at a minimum.

Five hundred images were then acquired at 200 ms intervals using imaging parameters that led to minimal acquisition photobleaching. Signal intensities were measured using ImageJ at bleached, nuclear, and background regions and min-max normalized (Ellenberg et al, 1997). Data were analyzed using Berkeley Madonna.

For Phf2-GFP, a sum of two exponential functions to represent nuclear diffusing and chromatin-bound populations was chosen and fit the data well. The chosen model has previously been applied to other chromatin-binding proteins and was shown to be in agreement with more direct chromatin binding measurements such as single-molecule tracking, a technique that requires a custom microscope setup (Mazza et al, 2012).

For Smc1-GFP FRAP analysis, a sum of three exponential functions was chosen as a model to represent nuclear diffusing, transiently chromatin-associated, and chromatin-bound populations. These populations were identified in previous FRAP experiments from our group by analyzing unbound cohesin in mitosis and cohesin ATPase mutants that are incapable of chromatin entrapment (Ladurner et al, 2014).

## RNA interference

Transfection of mouse Phf2 siRNA (Dharmacon) and Scc1 siRNA (Ambion, ID s72658) was performed using Lipofectamine RNAi-MAX (Invitrogen) according to the manufacturer's instructions. All siRNAs were used at a final concentration of 100 nM.

## Protein purification from Sf9 cells and in vitro binding assay

His-tagged full-length Phf2 and mutants were cloned into the baculovirus expression vector pLIB (Invitrogen) and transfected into Sf9 cells using the Bac-to-Bac Baculovirus Expression System (Invitrogen). Phf2 and truncated mutants was purified with high-salt buffer (25 mM KPhosphate buffer, pH 7.5, 500 mM KCl, 2 mM $MgCl_2$, 5% glycerol) on Ni-NTA agarose (Qiagen) and eluted with imidazole (Sigma). Cohesin complexes were expressed and purified via Ni-NTA agarose, followed by anti-FLAG M2 agarose (Sigma), as described (Huis in't Veld et al, 2014). The in vitro binding assay was performed by mixing FLAG-beads binding cohesin complexes with purified Phf2 at 4 °C for 1 h in binding buffer (35 mM Hepes, pH 7.5, 100 mM NaCl, 5% glycerol, 0.01% Tween). The beads were washed three times with binding buffer, eluted with FLAG peptide, and analyzed by silver staining or Western blot.

## ChIP followed by next-generation sequencing (ChIP-Seq)

ChIP was performed as described (Wendt et al, 2008). One 24 cm[2] tray was used per ChIP experiment. Cells were cross-linked with 1/10th medium volume of crosslinking solution (11% formaldehyde, 100 mM NaCl, 0.5 mM EGTA, 50 mM Hepes, pH 8.0) at room temperature for 10 min and subsequently quenched with 125 mM glycine for 5 min. Cells were washed with PBS, collected by scraping, and centrifuged. Cell pellets were lysed in lysis buffer (50 mM Tris-HCl, pH 8.0, 10 mM EDTA, pH 8.0, 1% SDS, protease inhibitors) on ice for 20 min. DNA was sonicated for 12 cycles (30 s on/off) using a Bioruptor. Five volumes of dilution buffer (20 mM Tris-HCl, pH 8.0, 2 mM EDTA, pH 8.0, 1% Triton X-100, 150 mM NaCl, 1 mM PMSF) were added to the lysate, followed by pre-clearing with 100 μl Affi-Prep Protein A beads at 4 °C. Immuno-precipitation was performed with rabbit IgG or antibody overnight, followed by a 3-h incubation with Affi-Prep Protein A beads. Beads were washed twice with Wash Buffer 1 (20 mM Tris-HCl, pH 8.0, 2 mM EDTA, pH 8.0, 1% Triton X-100, 150 mM NaCl, 0.1% SDS, 1 mM PMSF), twice with Wash Buffer 2 (20 mM Tris-HCl, pH 8.0, 2 mM EDTA, pH 8.0, 1% Triton X-100, 500 mM NaCl, 0.1% SDS, 1 mM PMSF), twice with Wash Buffer 3 (10 mM Tris-HCl, pH 8.0, 2 mM EDTA, pH 8.0, 250 mM LiCl, 0.5% NP-40, 0.5% deoxycho-late), and twice with TE buffer (10 mM Tris-HCl, pH 8.0, 1 mM EDTA, pH 8.0). The samples were eluted twice with 200 μl elution buffer (25 mM Tris-HCl, pH 7.5, 5 mM EDTA, 0.5% SDS) by shaking at 65 °C for 20 min. The eluates were treated with RNase A

at 37 °C for 1 h and proteinase K at 65 °C overnight. Following the addition of 1 µl glycogen (20 mg/ml) and 1/10th volume sodium acetate (3 M, pH 5.2), DNA was extracted with phenol/chloroform/isoamyl alcohol (25:24:1) and precipitated with ethanol. The DNA was resuspended in $H_2O$ and submitted for library preparation and Illumina deep sequencing at the Campus Science Support Facility. From two to four biological replicates were used per samples, and the analysis was performed using the merged data (for more info on the single files please refer to the GEO repository).

## ChIP-Seq analysis

ChIP-Seq files were aligned to mm9 reference genome with Bowtie2. The peaks were called using MACS2 and the visualized tracks were normalized using SPMR. We visualized the tracks using IGV. To analyze the overlap between peaks and visualize it as Venn diagram we used pybedtools (https://daler.github.io/pybedtools/) and matplotlib (https://matplotlib.org/). To create heatmap plots and summary plots we used deeptools (https://deeptools.readthedocs.io/en/develop/). To calculate the ChIP-Seq reads drop on a group of peaks we used pyBigWig (https://github.com/deeptools/pyBigWig). To visualize the overlap between peaks of H3K4me3 in the indicated conditions as triangular heatmap we used intervene pairwise (https://intervene.readthedocs.io).

RNA-Seq RNA-Seq was performed as previously described (Busslinger et al, 2017). Total RNA was extracted using TRIzol (Invitrogen), and its purity was assessed with an Agilent RNA 6000 Nano kit (Agilent). mRNA was purified in two rounds using a Dynabeads mRNA purification kit (Invitrogen). The purified mRNA was fragmented in fragmentation buffer (8 mM Tris, pH 8.2, 50 mM Kac, and 4.5 mM MgAc) and further purified using an RNeasy column. The quality of mRNA isolation and fragmentation was analyzed with an Agilent RNA 6000 Pico kit (Agilent). The fragmented mRNA was used as a template for first-strand cDNA synthesis using random hexamers and the Superscript III First-Strand Synthesis kit (Invitrogen), followed by purification on a Mini Quick Spin column (Roche). Second-strand cDNA synthesis was performed using random hexamers and 200 µM dATP, dCTP, dGTP, and dUTP in the presence of RNase H, *Escherichia coli* DNA polymerase I, and DNA ligase (Invitrogen). The cDNA samples were purified with a MinElute Reaction Cleanup kit (Qiagen), and >5 ng of cDNA was submitted for library preparation and Illumina deep sequencing.

## RNA-Seq analysis

RNA sequencing data were derived through an Illumina HiSeq 2500v4 system in single-read (50 bp) mode. Quality and adapter trimming was performed using Trim Galore. Abundant sequence fractions (rRNA) have been removed using bowtie2 (Langmead and Salzberg, 2012). In such a way cleaned raw reads were mapped against the reference genome MGSCv37 (i.e., mm9) using STAR (reverse_stranded) (Dobin et al, 2013). Mapped reads are assigned to corresponding genes using featureCounts (Liao et al, 2014). Analysis of differentially expressed genes is performed using DESeq2 (Love et al, 2014). The mentioned algorithms as well as others are compiled in an RNA-Seq pipeline adapted by IMP/IMBA Bioinformatics Core Facility, it is based on the nf-core/rnaseq

pipeline [https://doi.org/10.5281/zenodo.1400710] and is built with nextflow (Ewels et al, 2020).

## Hi-C in MEFs

Hi-C was performed as described previously (Banigan et al, 2023). Briefly, $30 \times 10^6$ cells were cross-linked in 2% formaldehyde for 10 min and quenched with ice-cold glycine (0.125 M final concentration). Cells were snap-frozen and stored at −80 °C before cell lysis. Cells were lysed for 30 min in ice-cold lysis buffer (10 mM Tris-HCl, pH 7.5, 10 mM NaCl, 5 mM $MgCl_2$, 0.1 mM EGTA, and 0.2% NP-40) in the presence of protease inhibitors. Chromatin was solubilized in 0.6% SDS at 37 °C for 2 h and quenched by 3.3% Triton X-100. Chromatin was digested with 400 units of HindIII overnight at 37 °C. Fill-in of digested overhangs by DNA polymerase I, large Klenow fragment in the presence of 250 nM biotin-14-dATP for 90 min was performed prior to 1% SDS-based enzyme inactivation and dilute ligation with T4DNA ligase for 4 h at 16 °C. Cross-links of ligated chromatin were reversed overnight by 1% proteinase K incubation at 65 °C. DNA was isolated with 1:1 phenol:chloroform, followed by 30 min of RNase A incubation. Biotin was removed from unligated ends by incubation with 15 units of T4 DNA polymerase. DNA was sheared using an E220 evolution sonicator (Covaris, E220) and size selected to 150 to 350 bp by using AMPure XP beads. After end repair in a mixture of T4 polynucleotide kinase, T4 DNA polymerase, and DNA polymerase I, large (Klenow) fragment at room temperature for 30 min, dATP was added to blunted ends polymerase I, large fragment (Klenow $3' \rightarrow 5'$ exo-) at 37 °C for 30 min. Biotinylated DNA was collected by incubation in the presence of 10 µL of streptavidin-coated MyOne C1 beads, and Illumina paired-end adapters were added by ligation with T4 DNA ligase for 2 h at room temperature. A PCR titration (primers PE1.0 and PE2.0) was performed prior to a production PCR to determine the minimal number of PCR cycles needed to generate a Hi-C library. Primers were separated from the library using AMPure XP size selection prior 150 bp paired-end reads sequencing using illumina. Hi-C libraries were generated as described in (Wutz et al, 2017), with modifications as described below. $3 \times 10^7$ MEFs were fixed in 2% formaldehyde for 10 min, after which the reaction was quenched with ice-cold glycine (0.125 M final concentration). Cells were collected by centrifugation ($400 \times g$ for 10 min at 4 °C) and washed once with 50 ml PBS, pH 7.4. After another centrifugation step ($400 \times g$ for 10 min at 4 °C), the supernatant was completely removed and the cell pellets were immediately frozen in liquid nitrogen and stored at −80 °C. After thawing, the cell pellets were incubated in 50 ml ice-cold lysis buffer (10 mM Tris-HCl, pH 7.5, 10 mM NaCl, 5 mM $MgCl_2$, 0.1 mM EGTA, and 0.2% NP-40) for 1 h on ice. After centrifugation to pellet the cell nuclei ($400 \times g$ for 5 min at 4 °C), nuclei were washed twice with 1.2× NEBuffer 2 (New England BioLabs) and transferred to 1.5 ml Eppendorf tubes. The nuclei were then collected by centrifugation step ($400 \times g$ for 5 min at 4 °C) and a resuspended in 450 µl 1.2× NEBuffer 2 (New England BioLabs) with 13.5 µl of 20% SDS (0.6% final concentration), and the nuclei were incubated at 37 °C for 2 h with agitation (900 rpm). Triton X-100 was added to a final concentration of 3.3%, and the nuclei were incubated at 37 °C for 2 h with agitation (900 rpm). HindIII (New England BioLabs; 1500 units per 7 million

cells) restriction digestion was performed overnight at 37 °C with agitation (900 rpm). Using biotin-14-dATP (Life Technologies), dCTP, dGTP, and dTTP (Life Technologies; all at a final concentration of 30 μM), the HindIII restriction sites were then filled in with Klenow (New England BioLabs) for 1 h at 37 °C with shaking (700 rpm) for 10 s every 30 s. The nuclei were washed twice with ligation buffer and the ligation was performed for 12 h at 16 °C (2000 units T4 DNA ligase, Thermo Scientific) in a total volume of 100 μl ligation buffer (50 mM Tris-HCl, 10 mM $MgCl_2$, 1 mM ATP, 10 mM DTT, 100 μg/ml BSA, and 0.9% Triton X-100). After ligation, crosslinking was reversed by incubation with proteinase K (40 μl of 10 mg/ml in 300 μl Tris-EDTA buffer [TE]) at 65 °C overnight. An additional proteinase K incubation (65 μl of 10 mg/ml per 7 million cells starting material) at 65 °C for 2 h was followed by RNase A (Roche; 15 μl of 10 mg/ml per 7 million cells starting material) treatment and two sequential phenol/chloroform (Sigma) extractions. DNA precipitation was performed overnight at −20 °C (3 M sodium acetate, pH 5.2 [1/10 volume] and ethanol [2.5 volumes]), and the DNA was then spun down ($3200 \times g$ for 30 min at 4 °C). The pellets were resuspended in 400 μl TE (10 mM Tris-HCl, pH 8.0 and 0.1 mM EDTA) and transferred to 1.5-ml Eppendorf tubes. After another phenol/chloroform (Sigma) extraction and DNA precipitation overnight at −20 °C, the pellets were washed three times with 70% ethanol, and the DNA concentration was determined using Quant-iT Pico Green (Life Technologies). To remove biotin from non-ligated fragment ends, 30–40 μg of Hi-C library DNA was incubated with T4 DNA polymerase (New England BioLabs) for 4 h at 20 °C, followed by phenol/chloroform purification and DNA precipitation overnight at −20 °C. After a wash with 70% ethanol, sonication was carried out to generate DNA fragments with a size peak around 400 bp (Covaris E220 settings: duty factor, 10%; peak incident power, 140 W; cycles per burst, 200; time, 55 s). After end repair (T4 DNA polymerase, T4 DNA polynucleotide kinase, Klenow [all New England BioLabs] in the presence of dNTPs in ligation buffer [New England BioLabs]) for 30 min at RT, the DNA was purified (Qiagen PCR purification kit). dATP was added with Klenow exo- (New England BioLabs) for 30 min at 37 °C, after which the enzyme was heat inactivated (20 min at 65 °C). A double-size selection using AMPure XP beads (Beckman Coulter) was performed: First, the ratio of AMPure XP beads solution volume to DNA sample volume was adjusted to 0.6:1. After incubation for 15 min at RT, the sample was transferred to a magnetic separator (DynaMag-2 magnet; Life Technologies), and the supernatant was transferred to a new Eppendorf tube, while the beads were discarded. The ratio of AMPure XP beads solution volume to DNA sample volume was then adjusted to 0.9:1 final. After incubation for 15 min at RT, the sample was transferred to a magnet (DynaMag-2 magnet; Life Technologies). Following two washes with 70% ethanol, the DNA was eluted in 100 μl of TLE (10 mM Tris-HCl, pH 8.0 and 0.1 mM EDTA). Biotinylated ligation products were isolated using MyOne Streptavidin C1 Dynabeads (Life Technologies) on a DynaMag-2 magnet (Life Technologies) in binding buffer (5 mM Tris, pH 8, 0.5 mM EDTA, and 1 M NaCl) for 30 min at RT. After two washes in binding buffer and one wash in ligation buffer (New England BioLabs), PE adapters (Illumina) were ligated onto Hi-C ligation products bound to streptavidin beads for 2 h at RT (T4 DNA ligase NEB, in ligation buffer, and slowly rotating). After washing twice with wash buffer (5 mM Tris, 0.5 mM EDTA, 1 M NaCl, and 0.05% Tween-20) and then

once with binding buffer, the DNA-bound beads were resuspended in a final volume of 90 μl NEBuffer 2. Bead-bound Hi-C DNA was amplified with seven PCR amplification cycles (36–40 individual PCRs) using PE PCR 1.0 and PE PCR 2.0 primers (Illumina). After PCR amplification, the Hi-C libraries were purified with AMPure XP beads (Beckman Coulter). The concentration of the Hi-C libraries was determined by Bioanalyzer profiles (Agilent Technologies) and qPCR (Kapa Biosystems), and the Hi-C libraries were paired-end sequenced (HiSeqv4, Illumina) at VBCF NGS.

### Hi-C analysis

Illumina sequencing was performed on all Hi-C libraries with 150 bp paired-end reads. Two replicate data sets for each library were truncated, filtered, and aligned against the human genome assembly hg19 (GRCh37) using bowtie2 and the HiCUP processing pipeline version 0.7.4, and finally merged (Wingett et al, 2015), HiCUP:pipeline for mapping and processing Hi-C data F1000Research, 4:1310, https://doi.org/10.12688/f1000research.7334.1). Alignments were converted into input for juicer_tools pre (juicer tools version 1.22.016; as well as input for HOMER version 4.11 (http://homer.ucsd.edu/homer/.98 PMID: 20513432). All juicer-based contact matrices used for analysis were Knight-Ruiz normalized. Matrix maps of Hi-C were visualized using Juicebox (https://github.com/aidenlab/Juicebox). Contact frequencies were calculated and plotted by cooltools using the open2c-examples jupyter notebook (https://github.com/open2c/open2c_examples). Loop annotation, also known as corner peak annotation, and merging was performed using 'juicer_tools hiccups' at default resolution. To overlap loops between genotypes, all the loops were resized to 50 kb × 50 kb and the overlap was calculated with pybedtools and plotted with matplotlib. TAD annotation and insulation scores were generated by cooltools using the open2c-examples jupyter notebook (https://github.com/open2c/open2c_examples). TAD boundaries were calculated using cool files at 10 kb resolution with a moving window of 100 kb. Strong boundaries were defined as having boundary strength values bigger than 0.7 (Appendix Fig. S6D). The overlap between TAD boundaries was calculated using genomic coordinates +/−20 kb. ChIP-seq summary plots on TADs boundaries were plotted using pyBigWig (https://github.com/deeptools/pyBigWig) and matplotlib (https://matplotlib.org/). Two biological replicates per sample were merged and analyzed in the manuscript.

## Data availability

All study data are included in the article and/or supporting information. ChIP-seq data are available in the Gene Expression Omnibus (GSE278142); Hi-C (GSE278143); RNA-seq (GSE281657). The mass spectrometry proteomics data have been deposited to the ProteomeXchange Consortium via the PRIDE repository with the dataset identifier PXD057926). Raw files for making the figures were deposited to https://www.ebi.ac.uk/biostudies/studies/S-BSST1714.

The source data of this paper are collected in the following database record: biostudies:S-SCDT-10_1038-S44318-024-00348-2.

# Peer review information

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

## Acknowledgements

We are grateful to Yuuki Imai (Ehime University) for providing Phf2 conditional knockout mice, to Xiaobing Shi (Van Andel Institute), Ina Poser and Tony Hyman (Max Planck Institute for Cell Biology and Genetics) for providing plasmids, Antonio Tedeschi and Maria Helena Idarraga-Amado for providing Wapl conditional knockout mice, Benedikt Bauer and Daniela Götz for providing purified cohesin, David Cisneros for help with fluorescence microscopy, Anton Goloborodko (IMBA) for help and discussion, and all members of the Peters laboratory for discussions and support. We thank the IMP/IMBA/GMI BioOptics and Molecular Biology Service, and the VBCF Next-Generation Sequencing team for expert technical support. The work of Lorenzo Costantino was funded by the Austrian Academy of Sciences (ÖAW). The work in the Mechtler lab was supported by the infrastructure funding 4th call 2022/01 (AT-SCP) of the Austrian Research Promotion Agency (FFG) and further funded by the P35045-B project (Grant https://doi.org/10.55776/P35045) and the F 8801-B Meiosis project (Grant https://doi.org/10.55776/F88) of the Austrian Science Fund (FWF). Research in the laboratory of Jan-Michael Peters is supported by Boehringer Ingelheim, the Austrian Life Sciences Programme 2023 (LS23 IF, project FO999902549), the European Research Council under the European Union's Horizon 2020 Research and Innovation Programme (101020558), the Human Frontier Science Program (RGP0057/2018), and the Vienna Science and Technology Fund (LS19-029). J-MP is also an adjunct professor at the Medical University of Vienna.

## Author contributions

**Wen Tang**: Conceptualization; Data curation; Formal analysis; Investigation; Methodology; Writing—original draft; Writing—review and editing. **Lorenzo Costantino**: Conceptualization; Data curation; Formal analysis; Funding acquisition; Investigation; Methodology; Writing—original draft; Writing—review and editing. **Roman Stocsits**: Data curation; Formal analysis; Methodology. **Gordana Wutz**: Investigation; Methodology. **Rene Ladurner**: Data curation; Formal analysis; Investigation; Methodology. **Otto Hudecz**: Formal analysis; Investigation; Methodology. **Karl Mechtler**: Funding acquisition; Investigation; Methodology. **Jan-Michael Peters**: Conceptualization; Data curation; Supervision; Funding acquisition; Investigation; Writing—original draft; Writing—review and editing.

Source data underlying figure panels in this paper may have individual authorship assigned. Where available, figure panel/source data authorship is listed in the following database record: biostudies:S-SCDT-10_1038-S44318-024-00348-2.

## Disclosure and competing interests statement

The authors declare no competing interests.

# Expanded View Figures

**Figure EV1.  Phf2 partially colocalizes with CTCF genome wide.**

(**A**) Immunoblot analysis of Phf2 in *Phf2* KO MEFs. MEFs with floxed alleles of Phf2 with or without ERCre (*Phf2* F/F, ERCre/+ or *Phf2* F/F, no ERCre) were treated with 4-OHT for the indicated days and whole-cell extracts were analyzed using the indicated antibodies. (**B**) Venn diagram illustrating the overlap between ChIP-seq peaks of CTCF and Phf2 in WT MEFs. (**C**) Pile-up heat maps and summary plots of ChIP-seq signals for Phf2 (in WT and *Phf2* KO), H3K4me3, Smc3, CTCF, and PolII Ser5-P at overlap groups indicated in (**B**). The Zoom-in panels show the indicated sub-groups of ChIP-seq signals at different color scales. (**D**) Summary plots of ChIP-seq signals obtained for Phf2 and Smc3 at oriented CTCF sites. The binding at the CTCF sequence on the + and the − strand was assessed in WT (left) and *Wapl* KO MEFs (right). Source data are available online for this figure.

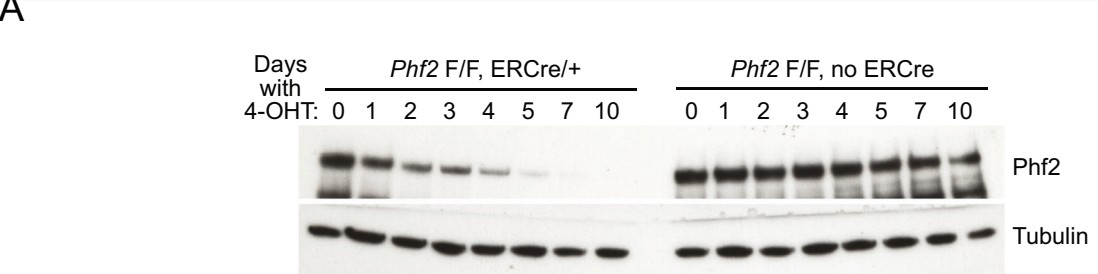

**A**

**B**

**C**

**D**

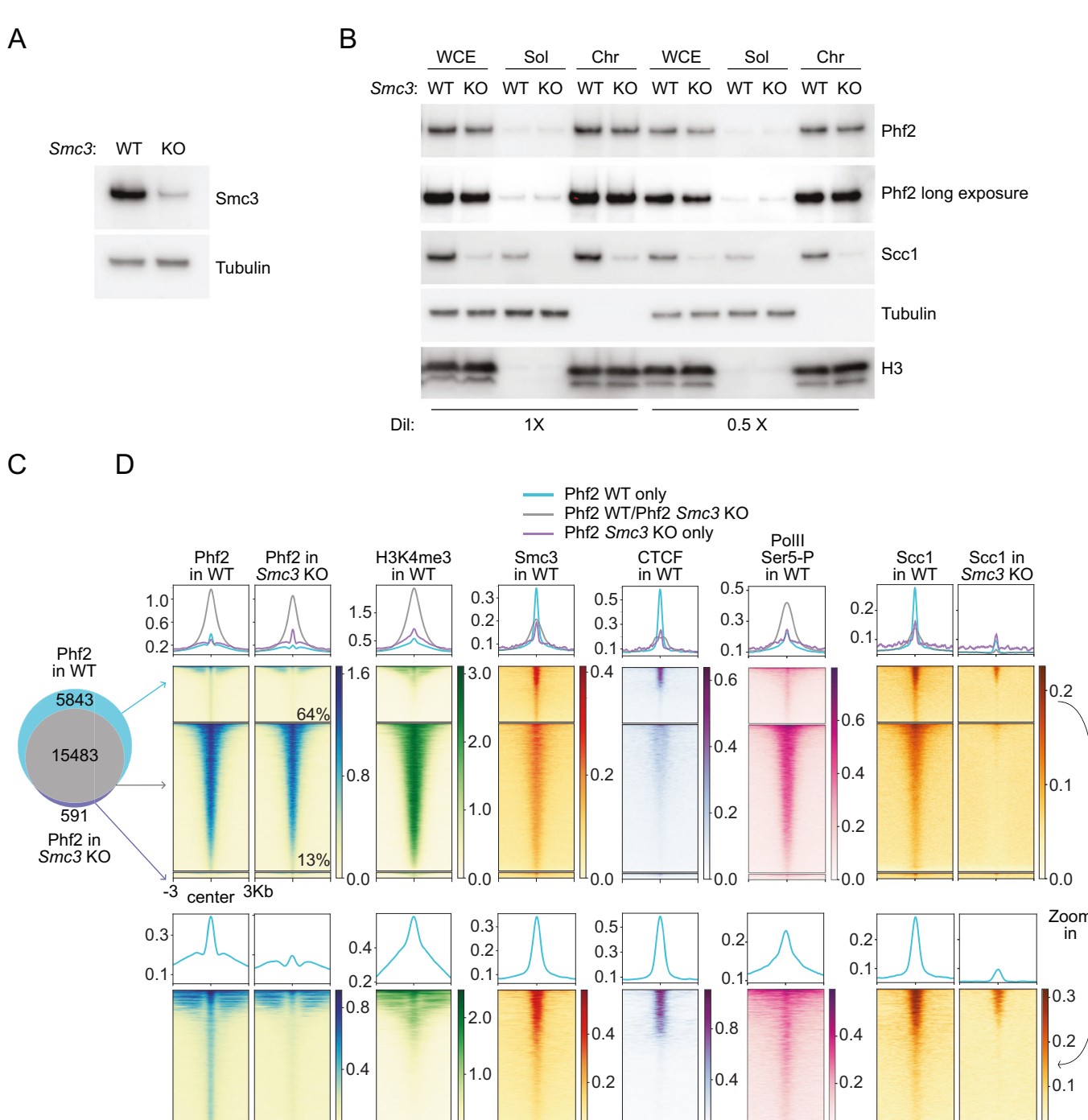

**Figure EV2. Enrichment of Phf2 at sites lacking H3K4me3 is dependent on cohesin.**

(A) Immunoblot analysis of whole-cell extracts from WT and *Smc3* KO MEFs using the indicated antibodies. (B) Immunoblot analysis of whole-cell extract (WCE), soluble (Sol), and chromatin-bound (Chr) fractions from WT and *Smc3* KO MEFs using the indicated antibodies. Two protein dilutions were assessed. Please note that a portion of this blot was used in Fig. 3A. (C) Venn diagram illustrating the overlap between ChIP-seq peaks of Phf2 in WT and *Smc3* KO MEFs. (D) Pile-up heat maps and summary plots of ChIP-seq signals for Phf2 (in WT and *Smc3* KO), H3K4me3, Smc3, CTCF, PolII Ser5-P, and Scc1 (in WT and *Smc3* KO; data from Busslinger et al, 2017) at overlap groups indicated in (C). Phf2 peaks present only in WT MEFs were reduced by 64% in read numbers upon Smc3 depletion, whereas Phf2 peaks that were present in WT and *Smc3* KO MEFs were reduced by only 13%. Zoom-in panels show the indicated sub-groups of ChIP-seq signals at different color scales. The panels from Fig. 3D are shown here again to facilitate a direct side-by-side comparison with the extended panels presented in this figure.

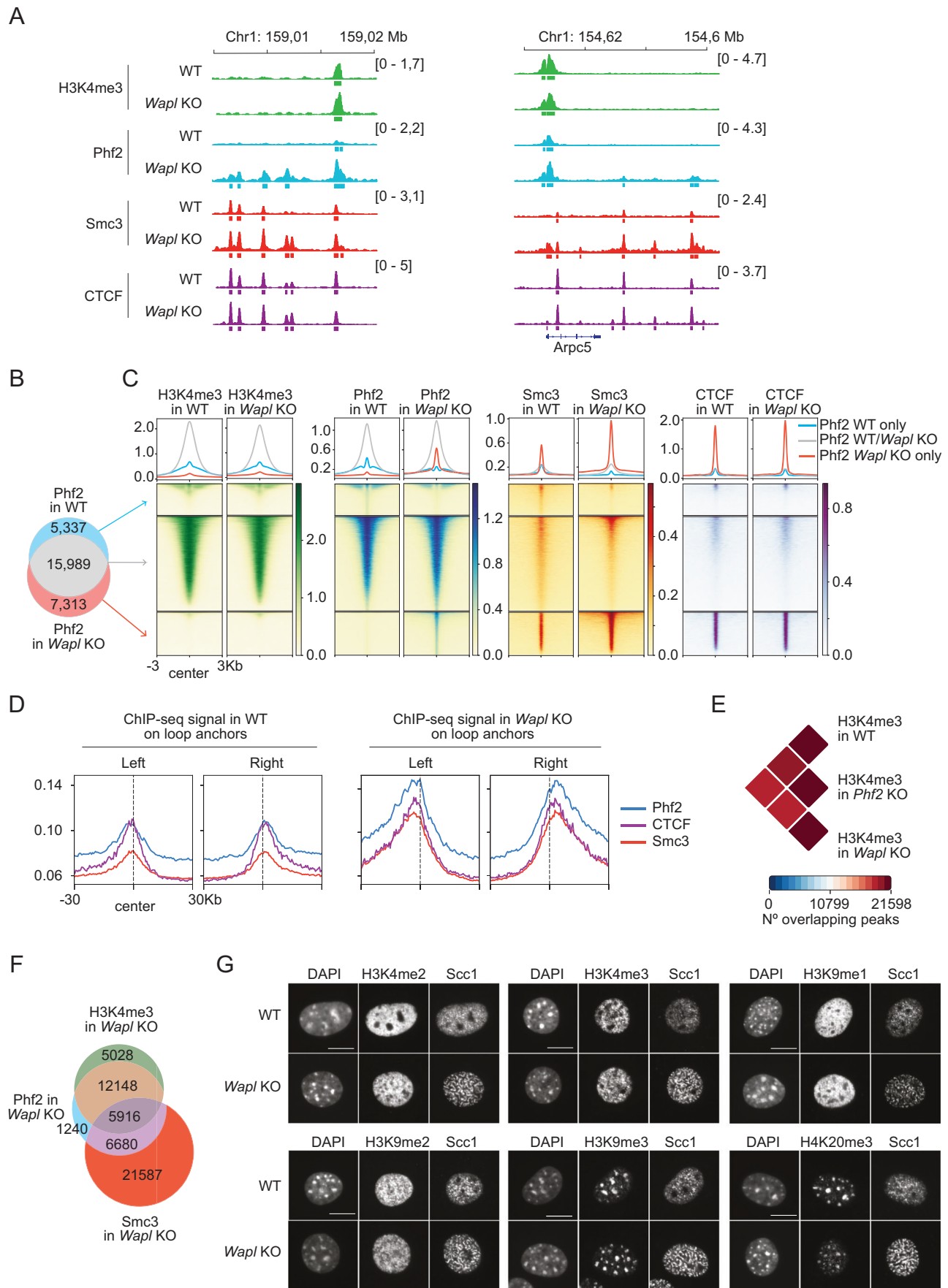

◀ **Figure EV3. Wapl depletion re-positions Phf2 with cohesin in the genome.**

(A) Binding of H3K4me3 (in WT and *Wapl* KO), Phf2 (in WT and *Wapl* KO), Smc3 (in WT and *Wapl* KO), and CTCF (in WT and *Wapl* KO) at two representative loci determined by ChIP-seq. The tracks from Fig. 5A are presented again in the left panel, now expanded to include all conditions. (B) Venn diagram showing the overlap between ChIP-seq peaks of Phf2 in WT and *Wapl* KO. (C) Pile-up heat maps and summary plots of ChIP-seq signals of H3K4me3 (in WT and *Wapl* KO), Phf2 (in WT and *Wapl* KO), Smc3 (in WT and *Wapl* KO), and CTCF (in WT and *Wapl* KO) at overlap groups indicated in (B). The panels from Fig. 5C are shown here again to facilitate a direct side-by-side comparison with the extended panels presented in this figure. (D) Summary plots of ChIP-seq signals obtained for Phf2, Smc3, and CTCF at loop anchors. The binding at left and right loop anchors was assessed in WT (left) and *Wapl* KO MEFs (right). (E) Triangular heatmap illustrating the overlap between ChIP-seq peaks of H3K4me3 in WT, *Phf2* KO, and *Wapl* KO. (F) Venn diagram showing the overlap between ChIP-seq peaks of Phf2, H3K4me3, and Smc3 in *Wapl* KO. (G) Fluorescence microscopy images for H3K4me2, H3K4me3, H3K9me1, H3K9me2, H3K9me3, H4K20me3, and Scc1 in WT and *Wapl* KO MEFs. Representative images of WT or *Wapl* KO MEFs stained with DAPI and antibodies to H3K4me2 (top-left), H3K4me3 (top-middle), H3K9me1 (top-right), H3K9me2 (bottom-left), H3K9me3 (bottom-middle), H4K20me3 (bottom-right) and Scc1. Scale bar, 10 μm. Source data are available online for this figure.

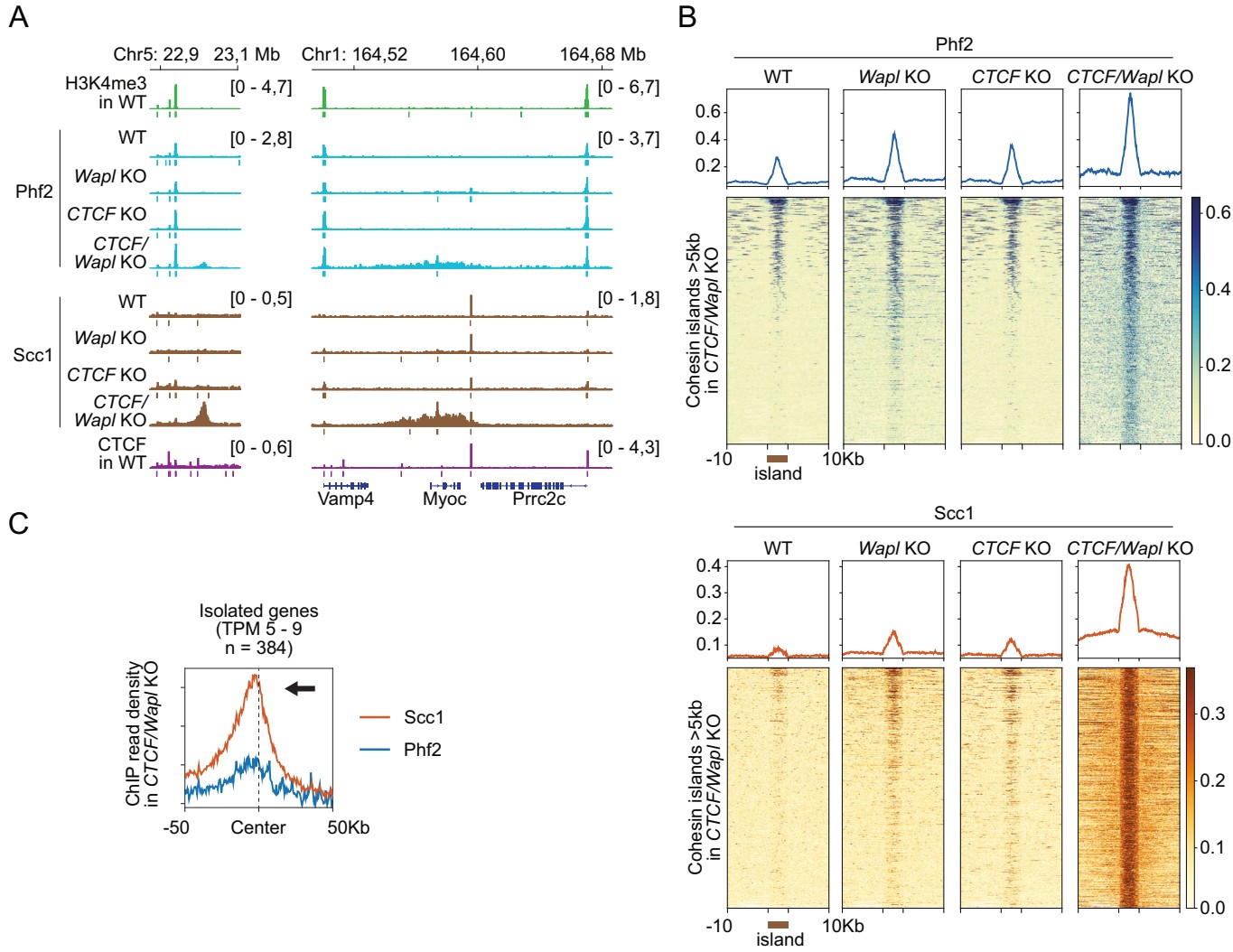

**Figure EV4. Wapl and CTCF depletion re-positions Phf2 with cohesin in the genome.**

(A) Binding of H3K4me3, Phf2 (in WT, *Wapl* KO, *CTCF* KO, and *CTCT/Wapl* KO MEFs), Scc1 (in WT, *Wapl* KO, *CTCF* KO, and *CTCT/Wapl* KO MEFs), and CTCF at two representative loci determined by ChIP-seq. The tracks from Fig. 5D are presented again in the right panel, now expanded to include all conditions. (B) Pile-up heat maps and summary plots of ChIP-seq signals obtained for Phf2 and Scc1 (in WT, *Wapl* KO, *CTCF* KO, and *CTCT/Wapl* KO MEFs) at cohesin islands found in *CTCF/Wapl* KO MEFs bigger than 5 kb (islands were stretched to equal size). The panels from Fig. 5E are shown here again to facilitate a direct side-by-side comparison with the extended panels presented in this figure. (C) Summary plots of ChIP-seq signals obtained for Scc1 and Phf2 in *CTCT/Wapl* KO MEFs for cohesin islands where transcription came only from one side, as measured by RNA-seq in Busslinger et al, 2017.

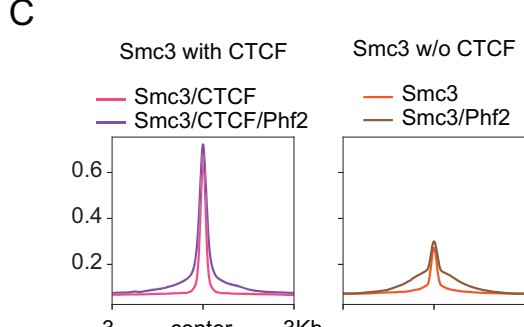

**Figure EV5. Phf2 localizes cohesin to TSSs in the absence of CTCF.**

(A) Venn diagram showing the overlap between ChIP-seq peaks obtained for Smc3 (in WT and *Phf2* KO), and Phf2 in WT. (B) Pile-up summary plots (top) and heat maps (bottom) of ChIP-seq signals obtained for H3K4me3 (in WT and *Phf2* KO), Phf2 (in WT and *Phf2* KO), Smc3 (in WT and *Phf2* KO), and CTCF at overlap groups indicated in (A). Numbers indicate the reduction in ChIP-seq signal of Smc3 peaks in *Phf2* KO MEFs at overlap groups indicated in (A). The Zoom-in panels show the indicated sub-groups of ChIP-seq signals at different color scales. The panels from Fig. 8C are shown here again to facilitate a direct side-by-side comparison with the extended panels presented in this figure. (C) Summary plots of ChIP-seq signals obtained for Smc3 at Smc3 peaks colocalizing with CTCF (left), or without CTCF (right). Peaks were further divided into colocalizing with Phf2 peaks or not.

