## [Peer Review File · The EMBO Journal]

Cohesin positions the epigenetic reader Phf2 within the genome

Wen Tang, Lorenzo Costantino, Roman Stocsits, Gordana Wutz, Rene Ladurner, Otto Hudecz, Karl Mechtler, and Jan-Michael Peters

Corresponding author(s): Jan-Michael Peters (Jan-Michael.Peters@imp.ac.at)

Review Timeline:

Submission Date:	13th Sep 24
Editorial Decision:	8th Nov 24
Revision Received:	18th Nov 24
Editorial Decision:	10th Dec 24
Revision Received:	11th Dec 24
Accepted:	12th Dec 24

Editor: Cornelius Schneider

Transaction Report:

Dear Dr Peters,

Thank you again for submitting your manuscript to the EMBO Journal, and for our productive meeting. As discussed during this meeting we would therefore like to invite you to revise your manuscript based on the concerns raised by all three referees, but we do not require an in-depth experimental investigation of the function of Phf2.

I should also add that it is The EMBO Journal policy to allow only a single major round of revision and that it is therefore important to resolve the main concerns at this stage.

We generally allow three months as standard revision time, which can be extended to 6 months in case of major revisions, such as the experiments required here. As a matter of policy, competing manuscripts published during this period will not negatively impact on our assessment of the conceptual advance presented by your study. However, we request that you contact the editor as soon as possible upon publication of any related work, to discuss how to proceed. Should you foresee a problem in meeting the deadline, please let us know in advance and we may be able to grant an extension.

Thank you for the opportunity to consider your work for publication. I look forward to your revision.

Yours sincerely,

Cornelius Schneider

Cornelius Schneider, PhD
Editor
The EMBO Journal
c.schneider@embojournal.org

We realize that it is difficult to revise to a specific deadline. In the interest of protecting the conceptual advance provided by the work, we recommend a revision within 3 months (6th Feb 2025). Please discuss the revision progress ahead of this time with the editor if you require more time to complete the revisions.

Referee #1:

In this manuscript, Jan-Michael Peters and colleagues describe a new interactor of the cohesin complex, Phf2. Not many cohesin interacting proteins are known. This is possibly due to the low affinity interactions between cohesin and other chromatin proteins or to the role of chromatin itself in stabilizing these interactions. To circumvent this problem, the authors leverage cross-linking-based strategies in conditions where cohesin is mostly bound to chromatin (WAPL KO) to identify new cohesin interacting proteins. Using a battery of ChIP-seq and Hi-C studies in WT and mutant conditions, the authors then characterize cohesin-dependent and independent mechanisms of Phf2 recruitment to chromatin and its corresponding functions.

Of particular interest is the observation that one of the regions within Phf2 that interacts with cohesin contains a YEY motif. As the authors also point out, there is a similar motif in CTCF (YxY motif) that regulates CTCF:cohesin interactions. This observation - which I think is one of the most important of this story - suggests an interesting competition between Phf2 and CTCF for cohesin and its activity on chromatin. I believe that is also the model that the authors would like to suggest in their Discussion and it is consistent with their data. This observation is important because it raises new and interesting hypotheses regarding the functional crosstalk between chromatin modifying enzymes (like Phf2) and architectural proteins (like CTCF and cohesin). For instance, are the interactions between most chromatin modifying enzymes and cohesin mutually exclusive with CTCF? And in cases when they are not, why is that? Given the relevance of this observation, I wonder if the authors could directly test it by mutating the YEY motif in Phf2.

Overall, this manuscript is important as it reveals a new functional link between loop-extrusion and chromatin proteins/enzymes.

Minor Comments:

1. In Figure 2C, the authors describe 4262 Phf2 binding sites where there is overlap between CTCF and Phf2. What the mechanism by which Phf2 and CTCF can co-occupy the same sites on chromatin? Does the orientation of CTCF matter for this co-occupancy? If so, is it possible that stalled cohesin at these sites is a consequence of Phf2 binding, rather than CTCF?
2. Figure 2G should reference overlap groups indicated in F, not D.
3. It is a bit difficult to assess the methodology of the FRAP experiment, considering that it is not a well-established technique to reliably quantify parameters such as fraction bound and residence time (PMID: 22844090; 22183594) Can the authors expand on it in their methods?
4. In Figure 5 and EV7C, the authors show a repositioning of Phf2 upon WAPL KO that appears dependent on cohesin but not H3K4me3. I suggest moving EV7C in the main figure as this data strongly supports the authors' model whereby cohesin functions as a "carrier" of the "cargo" Phf2.

Referee #2:

This paper presents a novel finding by identifying Phf2 as a cohesin-interacting factor, adding valuable insights into how cohesin and histone modification enzymes cooperate in eukaryotic genome regulation. The data supporting the co-localization of Phf2 and cohesin at transcription start sites (TSSs) and other genomic loci suggest a significant, previously unrecognized layer of genome regulation. Given its contributions to understanding the interplay between epigenetic readers and genome architecture, I believe this study has potential merit for publication in the EMBO Journal.

The quality of data presented is high, with convincing evidence of Phf2 and cohesin co-localization across the genome. The paper's central finding that Phf2 may function as an "epigenetic reader" with the ability to interact and co-localize with cohesin at active TSSs provides new insights into chromatin biology. This contributes substantially to understanding how cohesin might be

recruited or localized by non-CTCF factors, thereby expanding knowledge on genome architecture, which is a hot topic in the field now. The study proposes Phf2's role in recruiting cohesin to active transcription sites, potentially influencing chromatin compartmentalization and DNA loop structures. This has implications for genome organization, particularly in how B compartment size is limited in the presence of Phf2.

I believe my review underscores the study's value, with targeted recommendations for enhancing clarity and robustness. The paper's findings on the Phf2-cohesin interaction provide a strong foundation for publication, contingent on minor improvements in data presentation and reproducibility validation.

My major concerns are as follows.

1. Phf2's functional role is not so clear. While Phf2's interaction with cohesin is novel, additional insights into its mechanistic contributions to genome regulation would strengthen the paper's impact.
2. No reproducibility tests are mentioned for ChIP-seq or Hi-C data. Repeated experiments (e.g., ChIP-qPCR for ChIP-seq and additional Hi-C replicates) and sequencing read statistics should be included for data robustness. Additionally, depositing sequencing data in a public database is essential for data transparency.

Minor Points

1. In Figures 4B, 4C, and 4D, no information is provided regarding error bars, nor is the statistical significance of the results indicated. Adding these details would clarify the reliability of these findings.
2. The subtle differences observed in Hi-C data between wild-type and Phf2 knockout samples should be verified for reproducibility, as they may affect interpretations of the study's findings on chromatin compartment size changes.

Referee #3:

In this manuscript, Tang and colleagues identified Phf2, an epigenetic reader, as a cohesin-binding protein using cohesin affinity purification followed by mass spectrometry. They demonstrated that Phf2 localizes to H3K4me3 nucleosomes at active transcription start sites (TSS) alongside cohesin. Notably, the deletion of cohesin primarily reduces Phf2 binding in regions lacking H3K4me3 rather than those containing it. Furthermore, the simultaneous deletion of Wapl and CTCF alters the genomic binding patterns of both Phf2 and cohesin. Finally, the authors showed that Phf2 deletion impairs cohesin binding at TSS sites without CTCF, decreases the number of short cohesin loops at H3K4me3-nucleosomes, and extends the length of the heterochromatic B compartment.

Main comments:

1. Although the authors identified Phf2 as binding to the kleisin-STAG module of cohesin, it remains unclear which specific domain of cohesin Phf2 interacts with.
2. To elucidate the role of Phf2 in the regulation of cohesin, H3K4me3, CTCF, and PolII Ser5-P, the authors should perform ChIP-seq in the context of Phf2 deletion.
3. For the microscopy examination, the authors should include merged images to assess co-localization, particularly for Figs. 1B, EV2G-H, and EV7F. Additionally, the use of black-and-white images complicates the interpretation of the data.
4. Regarding Fig. 1G and Fig. EV2C-D, what is the size of His-Phf2? The authors should include Phf2 WT as a control in Fig. 1G. Furthermore, in Fig. EV2, why is one band missing in the Eluate Dimer compared to Fig. 1G? Which band corresponds to Phf2 in this figure?
5. The authors analyzed small Smc3 peaks from the ChIP-seq experiment in Fig. 8A; however, it is uncertain whether these small peaks represent genuine binding events or merely background noise.

Minor comments:

1. In Fig. 1D, the middle panel depicting Phf2 protein levels does not seem to support the conclusion that "Wapl depletion did not only enrich Phf2 in cohesin samples."
2. How many times of Western blots were performed in the manuscript? The authors should quantify the signal bands and indicate biological replicates in the figure legend.
3. Fig. 3 appears to present the same data as Fig. EV5D.
4. The authors should convert all bar graphs in Figs. 4C-D to dot plots for clarity.
5. In Fig. EV3A, why do the numbers for "Phf2 in WT" and "H3K4me3 in WT" differ between the top and bottom panels?
6. Check the reference style for "Cheng L, et al.," and "Rhodes J, et al.,".

Reply to the referee's comments:

We would like to thank all three referees for their insightful and constructive comments. We will address each of them below (the referee's comments are in black, our replies in blue).

Referee #1 (Report for Author)

In this manuscript, Jan-Michael Peters and colleagues describe a new interactor of the cohesin complex, Phf2. Not many cohesin interacting proteins are known. This is possibly due to the low affinity interactions between cohesin and other chromatin proteins or to the role of chromatin itself in stabilizing these interactions. To circumvent this problem, the authors leverage cross-linking-based strategies in conditions where cohesin is mostly bound to chromatin (WAPL KO) to identify new cohesin interacting proteins. Using a battery of ChIP-seq and Hi-C studies in WT and mutant conditions, the authors then characterize cohesin-dependent and independent mechanisms of Phf2 recruitment to chromatin and its corresponding functions.

Of particular interest is the observation that one of the regions within Phf2 that interacts with cohesin contains a YEY motif. As the authors also point out, there is a similar motif in CTCF (YxY motif) that regulates CTCF:cohesin interactions. This observation - which I think is one of the most important of this story - suggests an interesting competition between Phf2 and CTCF for cohesin and its activity on chromatin. I believe that is also the model that the authors would like to suggest in their Discussion and it is consistent with their data. This observation is important because it raises new and interesting hypotheses regarding the functional crosstalk between chromatin modifying enzymes (like Phf2) and architectural proteins (like CTCF and cohesin). For instance, are the interactions between most chromatin modifying enzymes and cohesin mutually exclusive with CTCF?

It is not clear whether Phf2 is a chromatin modifying enzyme, but if it was, it would be the first one. In ChIP-seq assays cohesin-Phf2 and cohesin-CTCF overlaps are seen, but it is unclear whether these interactions occur simultaneously in the same cells.

And in cases when they are not, why is that? Given the relevance of this observation, could the authors directly test it by mutating the YEY motif in Phf2?

As suggested by the referee, we have mutated the YEY motif in Phf2 to AEA (Phf2 Y673A Y675A), analogous to how the YxF motif in CTCF had been mutated to AxA (Li et al., Nature 2020). To address whether this mutation affects the ability of Phf2 to bind cohesin we have expressed the Phf2 Y673A Y675A mutant in MEFs, depleted Wapl and analyzed whether Phf2 still co-localizes with the cohesin subunit Smc3 in vermicelli. We also generated Phf2 mutants in which we either deleted 39 residues around the YEY motif (Phf2 Δ 673-711) or mutated a similar YVY motif (Phf2 Y869A Y871A), which is located in an unstructured region that does *not* bind cohesin (Δ 4 Fig. 1F-G). As is shown in the new Figure 1I panel, the Phf2 Y673A Y675A and Phf2- Δ 673-711 mutants are greatly reduced in their co-localization with cohesin in vermicelli, whereas mutation of the

YVY motif did not have such an effect. These results support the prediction that Phf2's YEY motif binds cohesin in the same pocket used for CTCF binding.

Overall, this manuscript is important as it reveals a new functional link between loop-extrusion and chromatin proteins/enzymes.

We thank the referee for the positive assessment of our manuscript.

Minor Comments:

1. In Figure 2C, the authors describe 4262 Phf2 binding sites where there is overlap between CTCF and Phf2. What the mechanism by which Phf2 and CTCF can co-occupy the same sites on chromatin?

We thank the referee for raising this question, which we did not explicitly address in our manuscript. The answer is that it is not clear from our ChIP-seq data whether Phf2 and CTCF really co-occupy the same sites on chromatin. This is for two reasons:

- Our ChIP-seq data are obtained from large cell populations. If two proteins appear to overlap in the genome, it is therefore not clear whether they simultaneously bind to the same genomic sites in the same cells, or whether one protein occupies this site in some cells and the other protein this site in other cells.
- Although Phf2 and CTCF ChIP-seq signals appear to overlap at the population level, the resolution of this technique is not high enough to determine whether they are indeed binding to overlapping sites. For example, although individual cohesin and CTCF peaks clearly appear to overlap, piling up their ChIP-seq signals genome wide reveals that the peaks of these signals of CTCF and cohesin are slightly shifted relative to each other (see for example, Nagy et al., BMC Genomics 2016). This suggests that at the single-molecule level these proteins might actually be located next to each other, a notion that is directly supported by single-molecule and cryo-EM studies (Davidson et al., Nature 2023; Zhang et al., Mol. Cell 2023). To test whether this might also be true for Phf2 and CTCF we have performed similar ChIP-seq pile-up analyses and found that Phf2 overlaps more with cohesin than with CTCF. These new results are shown in Fig. EV4D and indicate that Phf2 might co-occupy overlapping genomic sites with its binding partner cohesin but not with CTCF.

We have now also mentioned explicitly in the Discussion section that overlaps in Phf2 and CTCF ChIP-seq signals do not necessarily mean that these proteins are located at exactly the same sites in individual cells.

Does the orientation of CTCF matter for this co-occupancy?

We have tested this possibility in the same new analysis shown in Fig. EV4D. We found that the direction of the CTCF site did not affect the relative orientation of CTCF, cohesin and Phf2 to each other, i.e. both cohesin and Phf2 ChIP-seq peaks were always enriched on the N-terminal side of

CTCF. In other words, when the orientation of CTCF is changed, the relative position of cohesin and Phf2 to CTCF is not changed but the profiles for all three proteins are simply mirror-imaged.

If so, is it possible that stalled cohesin at these sites is a consequence of Phf2 binding, rather than CTCF?

For several reasons, we suspect that CTCF is the main cause of cohesin pausing at CTCF sites:

- In our experiments, Phf2 depletion did not prevent cohesin accumulation at CTCF sites (Figure 8A-C).
- Previous work has shown that CTCF is required for accumulation of cohesin at CTCF sites (Parelho et al., Cell 2008; Wendt et al., Nature 2008; Busslinger et al., Nature 2017).
- Single-molecule experiments have indicated that CTCF is sufficient to function as an orientation-dependent barrier to loop extruding cohesin (Davidson et al., Nature 2023; Zhang et al., Mol. Cell 2023).

However, our data indicate that Phf2 indeed contributes to the accumulation of cohesin at TSSs that are lacking CTCF (Figure 8).

2. Figure 2G should reference overlap groups indicated in F, not D.

We thank the reviewer for catching this mistake. We corrected the figure legend accordingly.

3. It is a bit difficult to assess the methodology of the FRAP experiment, considering that it is not a well-established technique to reliably quantify parameters such as fraction bound and residence time (PMID: 22844090; 22183594) Can the authors expand on it in their methods?

We have now expanded the description of the FRAP methodology and included references, as indicated below. FRAP analysis has been extensively used for assessing cohesin binding to DNA, and the measures are in accordance with other techniques. Moreover, we have now deposited the raw images and measures used to generate these figures in the EMBO database (biostudies) for users to consult.

“iMEFs (Wapl F/F, ERCCre/+) expressing Phf2/Smc1-GFP were imaged on a Zeiss LSM5 Duo confocal microscope using a 63× Plan-Apochromat objective and an open pinhole. Twenty images were acquired before bleaching a radial spot ($r = 2 \mu\text{m}$) three times at 100% laser intensity (100 mW diode 488). Bleaching with high laser power allowed for a short total bleach time, keeping intra-bleach diffusion effects at a minimum.

Five hundred images were then acquired at 200 ms intervals using imaging parameters that led to minimal acquisition photobleaching. Signal intensities were measured using ImageJ at bleached, nuclear, and background regions and min-max normalized (Ellenberg et al., 1997). Data were analyzed using Berkeley Madonna.

For Phf2-GFP, a sum of two exponential functions to represent nuclear diffusing and chromatin-bound populations was chosen and fit the data well. The chosen model has previously been applied to other chromatin-binding proteins and was shown to be in agreement with more direct

chromatin binding measurements such as single-molecule tracking, a technique that requires a custom microscope setup (Mazza et al. 2012).

For Smc1-GFP FRAP analysis, a sum of three exponential functions was chosen as a model to represent nuclear diffusing, transiently chromatin-associated, and chromatin-bound populations. These populations were identified in previous FRAP experiments from our group by analyzing unbound cohesin in mitosis and cohesin ATPase mutants that are incapable of chromatin entrapment (Ladurner, et al. 2014)."

4. In Figure 5 and EV7C, the authors show a repositioning of Phf2 upon WAPL KO that appears dependent on cohesin but not H3K4me3. I suggest moving EV7C in the main figure as this data strongly support the authors model whereby cohesin functions as a "carrier" of the "cargo" Phf2.

We agree with the reviewer that the results shown in Fig. EV7C corroborate our conclusion, but since the H3K4me3 distribution is not changing and because we are lacking space in the main figure we would like to leave the H3K4m3 data in EV7C.

Referee #2 (Report for Author)

This paper presents a novel finding by identifying Phf2 as a cohesin-interacting factor, adding valuable insights into how cohesin and histone modification enzymes cooperate in eukaryotic genome regulation. The data supporting the co-localization of Phf2 and cohesin at transcription start sites (TSSs) and other genomic loci suggest a significant, previously unrecognized layer of genome regulation. Given its contributions to understanding the interplay between epigenetic readers and genome architecture, I believe this study has potential merit for publication in the EMBO Journal.

The quality of data presented is high, with convincing evidence of Phf2 and cohesin co-localization across the genome. The paper's central finding that Phf2 may function as an "epigenetic reader" with the ability to interact and co-localize with cohesin at active TSSs provides new insights into chromatin biology. This contributes substantially to understanding how cohesin might be recruited or localized by non-CTCF factors, thereby expanding knowledge on genome architecture, which is a hot topic in the field now. The study proposes Phf2's role in recruiting cohesin to active transcription sites, potentially influencing chromatin compartmentalization and DNA loop structures. This has implications for genome organization, particularly in how B compartment size is limited in the presence of Phf2.

I believe my review underscores the study's value, with targeted recommendations for enhancing clarity and robustness. The paper's findings on the Phf2-cohesin interaction provide a strong foundation for publication, contingent on minor improvements in data presentation and reproducibility validation.

We thank the reviewer for the positive comments on our manuscript.

My major concerns are as follows.

1. Phf2's functional role is not so clear. While Phf2's interaction with cohesin is novel, additional insights into its mechanistic contributions to genome regulation would strengthen the paper's impact.

We fully agree with the reviewer that elucidating the mechanistic role of Phf2 in genome regulation will be important. However, we expect that this may take considerably more time and therefore feel that this would be beyond the scope of our current manuscript. For comparison, the role of the related fission yeast protein Epe1 in regulating heterochromatic B compartments has been discovered 17 years ago (Trewick et al., EMBO J., 2007) and has since then been studied by leading research groups in the epigenetics field (Robin Allshire, Shiv Grewal, Danesh Moazed, Sigurd Braun and others), yet the mechanistic basis of Epe1's function in heterochromatin regulation has not yet been elucidated yet.

2. No reproducibility tests are mentioned for ChIP-seq or Hi-C data. Repeated experiments (e.g., ChIP-qPCR for ChIP-seq and additional Hi-C replicates) and sequencing read statistics should be included for data robustness. Additionally, depositing sequencing data in a public database is essential for data transparency.

We thank the reviewer for pointing this out. All ChIP-seq and Hi-C results described in our manuscript are based on 2-4 biological replicates. We have now stated the number of replicates used for each experiment explicitly in the legends and the material and methods. All raw (single files) and processed data (merged data) have been deposited to GEO database and the accession numbers mentioned in the manuscript.

Minor Points

1. In Figures 4B, 4C, and 4D, no information is provided regarding error bars, nor is the statistical significance of the results indicated. Adding these details would clarify the reliability of these findings.

Error bars, statistical significance, and number of cells used are stated in the figures and legends that show and describe the FRAP experiments. The raw images and the measures used to generate these figures have also been deposited in the EMBO database (biostudies).

2. The subtle differences observed in Hi-C data between wild-type and Phf2 knockout samples should be verified for reproducibility, as they may affect interpretations of the study's findings on chromatin compartment size changes.

We performed two biological replicates per Hi-C sample, and the findings were confirmed in each replicate. To properly assess the magnitude of each phenotype, we used the merged data to maximize the number of reads available.

Referee #3 (Report for Author)

In this manuscript, Tang and colleagues identified Phf2, an epigenetic reader, as a cohesin-binding protein using cohesin affinity purification followed by mass spectrometry. They demonstrated that Phf2 localizes to H3K4me3 nucleosomes at active transcription start sites (TSS) alongside cohesin. Notably, the deletion of cohesin primarily reduces Phf2 binding in regions lacking H3K4me3 rather than those containing it. Furthermore, the simultaneous deletion of Wapl and CTCF alters the genomic binding patterns of both Phf2 and cohesin. Finally, the authors showed that Phf2 deletion impairs cohesin binding at TSS sites without CTCF, decreases the number of short cohesin loops at H3K4me3-nucleosomes, and extends the length of the heterochromatic B compartment.

Main comments:

1. Although the authors identified Phf2 as binding to the kleisin-STAG module of cohesin, it remains unclear which specific domain of cohesin Phf2 interacts with.

To address this question, we tested whether the YEY motif in Phf2 that aFold had predicted to bind to a conserved essential sequence (CES) on the STAG-kleisin module of cohesin is indeed required for cohesin binding (see also our reply to the first comment of reviewer 1).

To address whether this mutation affects the ability of Phf2 to bind cohesin we have expressed the resulting AEA mutant (Phf2 Y673A Y675A) in MEFs, depleted Wapl and analyzed whether Phf2 still co-localizes with the cohesin in vermicelli. We also generated Phf2 mutants in which we either deleted 39 residues around the YEY motif (673-711) or mutated a similar YVY motif, which is expected to *not* bind to cohesin ($\Delta 4$ in Fig. 1F-G). As is shown in the new Figure 1I panel, the Phf2 Y673A Y675A and Phf2-D673-711 mutants are greatly reduced in their co-localization with cohesin in vermicelli, whereas mutation of the YVY motif (Phf2 Y869A Y871A) did not have such an effect. These results confirm a key prediction of the alphaFold model and thus support the hypothesis that Phf2's YEY motif interacts with the CES of cohesin.

In the future it will be important to also obtain direct evidence for this interaction through biochemical and structural approaches. However, since these will be more time consuming, we feel that these are beyond the scope of our current study (please note that between the discovery of cohesin-CTCF interactions in 2008 and insight into how these proteins interact twelve years elapsed; Li et al., Nature 2020).

2. To elucidate the role of Phf2 in the regulation of cohesin, H3K4me3, CTCF, and PolII Ser5-P, the authors should perform CHIP-seq in the context of Phf2 deletion.

We have now included H3K4me3 CHIP-seq data obtained from *Phf2* KO MEFs in the new Fig.EV11B. These results show that depletion of Phf2 is not affecting the levels or genomic distribution of H3K4me3.

We have not analyzed CTCF and Pol-II-Ser5-P in *Phf2* KO MEFs because there are plausible reasons to believe that their distribution will not be affected by Phf2 depletion:

- It is well established that CTCF binding is determined by the presence of its consensus binding motif in the genome and the methylation state of this sequence. Unless Phf2 depletion would alter DNA methylation, for which there is no indication, one would therefore not expect CTCF binding to change. This assumption is also supported by the findings that Phf2 binds cohesin, and that cohesin depletion does not alter CTCF ChIP-seq profiles (Nora et al., Cell 2017; Wutz et al., 2017), and by the observation that *Phf2* KO mice (Okuno et al., Diabetes 2013) lack the phenotypes observed in *CTCF* KO mice (Heath et al, EMBO J. 2008 and subsequent studies).
- We found that Phf2 depletion causes hardly any transcriptional changes in MEFs (Fig. EV9A). We would therefore not expect Pol-II-Ser5-P ChIP-seq signals to change in these cells.

3. For the microscopy examination, the authors should include merged images to assess co-localization, particularly for Figs. 1B, EV2G-H, and EV7F. Additionally, the use of black-and-white images complicates the interpretation of the data.

We have now added panels showing merged channels for the figures shown in Figures 1B, 1I, and EV2G-H. However, we feel that it will be important to keep the original black-and-white images because readers with color vision weaknesses will not be able to interpret the merged images. For red-green weakness these readers represent 8% of the male and 0.5% of the female population (including authors on this manuscript).

4. Regarding Fig. 1G and Fig. EV2C-D, what is the size of His-Phf2?

Phf2's molecular mass is 120,000 Da, but its apparent mass as judged by SDS-PAGE is 160,000 Da (Fig. EV2C). This difference is only to a very small extent caused by the 10x-His-tag that is present on the recombinant version because this tag has a mass of only 2,000 Da.

The molecular mass markers in the gels shown in Fig. EV2C and 2D are aligned to allow direct comparison. We indicated this now in the figure legend and added arrows with a label in the figures.

The authors should include Phf2 WT as a control in Fig. 1G.

We did not include full-length Phf2 in this experiment since it would co-migrate with the Stag/Smc1 subunit of cohesin (as can be seen in Fig. EV2C-D) and would therefore not be visible by silver staining. However, we also do not think that the inclusion of full-length Phf2 in this experiment is essential since three of the four Phf2 mutants tested bind to cohesin and thus serve as internal positive controls. Furthermore, the binding of full-length Phf2 to cohesin has been analyzed in Figure 1E, in this case by Western blotting to detect Phf2 despite its co-migration with the Stag/Smc1 subunit of cohesin.

Furthermore, in Fig. EV2, why is one band missing in the Eluate Dimer compared to Fig. 1G? Which band corresponds to Phf2 in this figure?

We suspect there is a misunderstanding here since no band is missing in the panels shown in Fig. EV2.

To briefly explain: we first analyzed the binding of full-length Phf2 to dimeric, trimeric and tetrameric complexes of cohesin bound to antibody beads. Since Phf2 comigrates with Stag/Smc1, we could only use WB to detect Phf2 (Fig1E). The samples were also analyzed by silver staining in Fig. EV2D, to show the cohesin subunits (in the eluate, are now labeled in the figure). This experiment revealed that Phf2 binds best to tetrameric cohesin. We therefore used tetrameric cohesin to test which deletion mutants of Phf2 can still bind to tetrameric cohesin in Fig. 1G.

5. The authors analyzed small Smc3 peaks from the ChIP-seq experiment in Fig. 8A; however, it is uncertain whether these small peaks represent genuine binding events or merely background noise.

Like the reviewer, we were concerned about this possibility. For this reason, we analyzed Smc3 by ChIP-seq in WT and *Phf2* KO MEFs in four separate biological replicates. The results presented are the result of the merged data analyzed using the default parameters of the peak-calling tool MACS2, to not introduce biases. Based on these analyses, we are convinced that the small peaks detected in WT MEFs represent genuine Smc3 signals and not background.

Minor comments:

1. In Fig. 1D, the middle panel depicting Phf2 protein levels does not seem to support the conclusion that "Wapl depletion did not only enrich Phf2 in cohesin samples."

There might also be a misunderstanding here. In the experiment shown in Fig.1D, we isolated Phf2-GFP from WT and Wapl KO MEFs. For this reason, the levels of Phf2-GFP are comparable in the middle panel of Fig. 1D. However, we then analyzed how much Smc3 co-immunoprecipitates with Phf2-GFP. This revealed that more Smc3 co-immunoprecipitated with Phf2-GFP from Wapl depleted cells than from WT cells, which supports the conclusion of our paper.

2. How many times of Western blots were performed in the manuscript? The authors should quantify the signal bands and indicate biological replicates in the figure legend.

All Western blots shown in the manuscript are representative of at least two biological replicates. The original scans of the replicates are deposited now in the EMBO database for transparency.

3. Fig. 3 appears to present the same data as Fig. EV5D.

This is correct and intentional. Including all panels in Fig. 3 would have made this figure very large. We therefore decided to show some panels only in Fig. EV5D but included the panels from Fig. 3 also in Fig. EV5D to allow for a better side-by-side comparison. We have now explained this in the legend of Fig. EV5D.

5. In Fig. EV3A, why do the numbers for "Phf2 in WT" and "H3K4me3 in WT" differ between the top and bottom panels?

We thank the reviewer for noticing this mistake, which we have now corrected.

4. The authors should convert all bar graphs in Figs. 4C-D to dot plots for clarity.

We updated the figures with dot plots, as requested.

6. Check the reference style for "Cheng L, et al.," and "Rhodes J, et a.,".

We thank the reviewer for noticing these inconsistencies, which we have now corrected.

Dear Dr Peters,

Thank you for submitting a revised version of your manuscript. Your study has now been seen by all original referees, who find that their previous concerns have been addressed and now recommend publication of the manuscript. There remain only a few mainly editorial points that have to be addressed before I can extend formal acceptance of the manuscript:

- Please remove the blank pages in ms file
- Please add the following funding information in eJP: the Austrian Academy of Science (OAW); 2022/01 (AT-SCP) of the Austrian Research Promotion Agency (FFG) and further funded by the P35045-B project (Grant-DOI 10.55776/P35045) and the F 8801-B Meiosis project (Grant-DOI 10.55776/F88) of the Austrian Science Fund (FWF);
- On the abstract page of the manuscript, please include 4-5 general keyword terms to enhance searchability.
- Please include the references for materials and methods" in the main reference list "References"
- Please rename the Conflict of Interest section into "Disclosure and Competing Interests Statement", in accordance with our updated Guide to Authors (<https://www.embopress.org/competing-interests>)
- As we are switching from a free-text author contribution statement towards a more formal statement based on Contributor Role Taxonomy (CRediT) terms, please remove the present Author Contribution section and instead specify each author's contribution(s) directly in the Author Information page of our submission system during upload of the final manuscript. See <https://casrai.org/credit/> for more information.
- Please adjust the in-text callouts for individual figures and figure panels: e.g. 1l appears to be missing
- There are 11 EV figures but we only allow for up to 5 EV figures, and the others should be compiled in an Appendix PDF. DATASET EV LEGENDS: source file names, titles, legends and manuscript callouts all need to be updated to Dataset EV1-EV5
- APPENDIX FILE WITH ToC: The Appendix file containing the remaining EV figures needs to be in PDF format; extra EV figures should be compiled in Appendix PDF with title page containing Appendix for + ms title and ToC with page numbers of the listed items; nomenclature should be Appendix Figure Sx throughout ms and Appendix PDF
- Please provide the Reagent and Tools Table. For more information, please check <https://www.embopress.org/page/journal/14602075/authorguide#structuredmethods> and download the template for Reagent Table (attached for your convenience)
- Please save the source data in a scheme one figure/folder and then uploaded as .zip files. E.g. all the Source data files for figure 1 need to be saved in a single folder and this needs to be zipped and then uploaded as "SD figure 1.zip" file. For EV and/or appendix figures, ZIP together all source data.
- Please provide suggestions for a short 'blurb' text prefacing and summing up the conceptual aspect of the study in two sentences (max. 250 characters), followed by 3-5 one-sentence 'bullet points' with brief factual statements of key results of the paper; they will form the basis of an editor-written 'Synopsis' accompanying the online version of the article. Please also provide an altered synopsis image, making sure that the aspect ratio conforms to our website's format - it should be exactly 550 pixels wide and between 300-600 pixels high.
- Please correct the section order which should be: Title page - Abstract & Keywords - Introduction - Results - Discussion - Methods - Data Availability - Acknowledgements - Disclosure and Competing Interests Statement - References - Figure Legends - Table(s) - Expanded View Figure Legends.
- Cell reuse between Figure 1 and Figure EV1 not listed in the figure legend. Could you please either replace the cell in question or explicitly mention this reuse in the figure legends
- Figure Legends (main + EV):

1. Please define the annotated p values ****/***/**/* as well as provide the exact p-values for the same in the legend of figures 4C, EV6 E as appropriate.
2. Please indicate the statistical test used for data analysis in the legends of figures 1A, 4C, D; EV1 C, EV6 E, F; EV9 A.
3. Please indicate what */ **/ ***/ **** represents; if this represents p values, please indicate the statistical test used and the exact p value in the legends of figures 7D.
4. Please note that the box plots need to be defined in terms of minima, maxima, centre, bounds of box and whiskers, and percentile in the legends of figures 6E, 7D.
5. Please note that information related to n is missing in the legends of figures 1A; 4B, C, D; 6E, 7D; EV1 C, EV6 E, F, D; EV9 A.
6. Please note that the error bars are not defined in the legends of figures 4B-D; EV6 E, F, D.
7. Please note that the scale bar needs to be defined for figures 4A, EV2 G, H.

With best regards,

Cornelius Schneider

Cornelius Schneider, PhD
Editor
The EMBO Journal
c.schneider@embojournal.org

Please remember: Digital image enhancement is acceptable practice, as long as it accurately represents the original data and

conforms to community standards. If a figure has been subjected to significant electronic manipulation, this must be noted in the figure legend or in the 'Materials and Methods' section. The editors reserve the right to request original versions of figures and the original images that were used to assemble the figure.

We realize that it is difficult to revise to a specific deadline. In the interest of protecting the conceptual advance provided by the work, we recommend a revision within 3 months (10th Mar 2025). Please discuss the revision progress ahead of this time with the editor if you require more time to complete the revisions.

Referee #1:

The authors have addressed all my comments. I therefore recommend this manuscript for publications. Congratulations on this important work.

Referee #2:

I reluctantly agree with the authors' response that the function of Phf2 is an issue for the future. The fact that the function of the gene in other organisms is not clearly known suggests that this issue is a difficult one to tackle.

The other points raised have been satisfactorily addressed.

Referee #3:

Author generally addressed most of my concerns.

All editorial and formatting issues were resolved by the authors.

Dear Dr. Peters,

I am pleased to inform you that your manuscript has been accepted for publication in the EMBO Journal.

Yours sincerely,

Cornelius Schneider, PhD
Editor
The EMBO Journal
c.schneider@embojournal.org
